# The shadow of the family: Historical roots of social trust in Europe

**Maria Kravtsova** [1,2,3]*, **Aleksey Oshchepkov** [1,2], **Christian Welzel**[3]

**1** HSE University, Moscow, Russia, **2** Free University of Berlin, Berlin, Germany, **3** Leuphana University, Lüneburg, Germany

* mkrav@zedat.fu-berlin.de

## Abstract

This study provides new evidence on how historical patterns of household formation shape the present-day level of trust. We test two distinct features of historical family arrangements that might be harmful to trust towards out-groups: (a) family extendedness in terms of the number of household members, and (b) generational hierarchy and gender relations within the household. To conduct our study, we compiled a historical database that reflects family structure and socio-economic development, mostly in the 19th century. The analysis was performed on a sample of 94 historical subnational units within eight contemporary Western and Eastern European countries that participated in the Life in Transition Survey in 2010. We find that cohabitation of several generations within the historical family and power of older generations over the younger are detrimental for out-group trust today. By contrast, family extendedness *per se* was revealed to have no impact on trust.

**Data Availability Statement:** The link to the replication package is https://github.com/krav1978/The-Shadow-of-the-Family.

**Funding:** The article was prepared within the framework of the HSE University Basic Research

## 1. Introduction

"Family is the crystal of a society"

Victor Hugo (1802–1885)

In this article, we investigate the historical predictors of one of the most essential ingredients of the entire modernization complex–social trust. Trust is an asset of particular importance because it facilitates cooperation among individuals, thus elevating their capacity for collective action and the number and quality of proliferated common goods [1, 2]. To the extent to which interpersonal trust extends to people in general, the collective action/common good effects of trust cross beyond specific networks, all the way into society as a whole [3, 4].

In our paper we consider the most beneficial form of social trust–trust towards out-groups. While in-group trust fosters, above all, in-group solidarity and mutual loyalty, out-group trust is the decisive lubricant for a myriad of inter-human transactions that drive modernization, prosperity, democracy, and impartial government [2, 5–9].

Given the centrality of out-group trust for development and well-being, the question is: where does it come from? We search for its deep historical roots, contributing to the literature that links the studied countries' contemporary economic, political, and cultural traits to household organization principles that were already prevalent in pre-industrial times [10–16].

Program. The funders had no role in study design, data collection and analysis, decision to publish, or preparation of the manuscript.

**Competing interests:** The authors have declared that no competing interests exist.

Inspired by the pioneering work of Hajnal [17], the above cited scholars consider pre-industrial patterns of family to be a significant factor in the present-day functioning of society, from prosperity to democracy to good government. We extend this literature by looking at the historical effects of family on out-group trust as a common source of prosperity, democracy and government quality.

Our study focuses on Europe for three reasons. First is the greater availability of historic family data compared to the other parts of the world. Second, Europe offers rich comparative contrasts because it is one of the most culturally diverse regions of the world: on the Schwartz-, Minkov- and Inglehart-Welzel cultural maps of the world, the distances between European countries cover two thirds of global cross-cultural variation. The third reason is that Europe offers, nevertheless, a controlled environment in which there is no potentially confounding influence of European colonialism, as is found in most other parts of the world.

The main explanatory variables come from historical censuses of the 19th century or earlier, and they are designed to capture two important dimensions in pre-industrial family arrangements: family extendedness and hierarchical relations within the family. These dimensions define the different family types that occurred across historical Europe. Our family types should be disentangled from the related concept of "family ties" [18]. Family ties imply that individuals attach great importance to the family and are ready to sacrifice their personal interests for it. The theory is not unambiguous about the relationship between extendedness/nuclearity and family ties. Alesina and Giuliano suggest that extended families are correlated with strong family ties; however, Banfield's "amoral familism" [19] refers to nuclear families in Southern Italy. This question deserves a level of study that is beyond the scope of the current paper.

By family extendedness we mean all possible extensions beyond the nuclear family unit, defined as consisting of a married couple and their children under the age of 20 (e.g., adult children, (grand)parents of the both spouses, grandchildren, siblings, aunts, uncles, cousins, etc.). The term "extendedness" is used in its general meaning as the number of relatives in the household. It should not be confused with P. Laslett's [20] term "extended family unit", which refers to a household with single relatives who are not included in the family nucleus. We are primarily interested in the number of adult household members, while the number of children, linked with fertility, is less valuable for our research purposes. We measure family extendedness using a battery of indicators based on the number of people within the household. By in-family hierarchy we mean the power of older generations over the younger, i.e., the patriarchal seniority principle, and the power of men over women, which are measured through composite indices derived from individual census sheets. The outcome variable is the contemporary out-group trust index, derived from the 2010 Life in Transition Survey (LiTS) conducted by the European Bank of Reconstruction and Development (EBRD).

The existing literature [12, 21, 22] does not draw a clear distinction between the nuclearity/extendedness and power dimensions. Indeed, these two dimensions overlap to some extent. The nuclear family is, by definition, less hierarchical as younger generations live separately from older generations, considerably reducing parents' authority over their adult offspring. In the case of the extended family the situation is more ambiguous. Extended families may imply adult children remaining in their parental household upon marriage (vertical family extensions), as well as a number of lateral relatives (siblings, aunts, uncles, nephews, nieces, cousins) and their families, forming an extended household (horizontal family extensions). In the former case we could speak about generational hierarchy, while the latter case implies a simple two-generational household, as lateral relatives are of the same generations as the members of the family nucleus. Moreover, the co-residence of parents and their adult children does not necessarily mean the dominance of older generations over the younger. Often such households

are headed by representatives of the younger generations, while parents have a dependent status. Obviously, such households cannot be considered to be traditional patriarchal families.

In the present study we explicitly compare the effects of family extendedness and of intra-family power hierarchies. We seek to determine how these two features affect the current level of out-group trust. All types of extended families are more self-sufficient and can perform multiple functions due to a large number of relatives. Therefore, they have less need for interpersonal contacts beyond the kin group or associations incorporating several families. This constitutes an "isolative" component of family extendedness. Meanwhile, the "cooperative" component involves horizontal family extension, (i.e., additional relatives of the same generation and the same social status as nuclear family members) as well as non-kin household members like servants. According to Putnam [2], interaction with a large number of people of equal social status on a daily basis triggers strong reciprocity norms and trust formation. We also expect that regular interaction with non-relatives evokes greater out-group trust.

Hierarchical relations, by contrast, hamper people's ability to trust and cooperate with each other [2] because those in power can easily violate agreements without being punished. Power relations may be especially harmful for trust formation when they are experienced early on in childhood within the family. In the course of our empirical analysis we find that contemporary out-group trust is predicted by the generational hierarchy within the pre-industrial family, while family extendedness combining isolative and cooperative components does not impact trust.

Our next contribution consists of a huge data collection at the subnational regional level, based on national censuses (some of which are available from Mosaic and IPUMS projects), which are the most accurate and comprehensive source of information. Aggregation at the level of subnational regions makes more sense than nation-level aggregation due to historically shifting country borders. Our core database contains historical indicators available for 94 regional units within eight contemporary countries: Albania, Croatia, France, Hungary, Romania, Slovakia, Sweden, and the United Kingdom (which correspond to seven historical states: Albania, France, England and Wales, the Hungarian part of the Habsburg Empire, Romania (Wallachia), Scotland, and Sweden).

The rest of our article is organized as follows. Section Two reviews the existing literature, discusses the persistence of the family effect on trust and formulates our hypotheses. Section Three describes the data, variables and methods. Section Four demonstrates the main findings, while the final section presents our conclusions. Descriptive statistics and supportive supplementary material are placed in the appendix.

## 2. Literature and hypotheses

### 2.1 Literature

There is a large body of research which acknowledges the distinctive persistence of trust and looks for its historical roots. The contemporary level of trust was linked to many historical factors including, for instance, the experience of free communes in medieval Italy [2, 23], the slave trade in West Africa 400 years ago [24], and the cross-country variation in non-despotic political institutions in the distant past [25].

We rely on the evidence that contemporary levels of trust might be predicted not only by the societal-level historical institutions but also by grass-root institutions, namely, families. Several papers relate historical family arrangements to social trust today. Schulz et al. [15], exploring variation in cousin-marriage across counties, European regions and ethnicities, finds that these practices, both in the past and in the present, are detrimental for generalized trust as well as for political participation and democracy. Furthermore, the cousin-marriage

tradition is beneficial for corruption [26]. Schulz et al. [15] come to the general conclusion that the dissolution of extended kin networks due to the prohibition of cousin-marriage by the Western Church explains why only a small number of societies could today be characterized as Western, Educated, Industrialized, Rich, and Democratic (WEIRD). Alesina & Giuliano [27] point in the same direction, suggesting that strong family ties, as measured using the World Values Survey (WVS), have a negative effect on generalized trust and political participation. Duranton, Rodríguez-Pose & Sandall [10] show that the nuclear family based on unigeniture is the most beneficial family type for trust formation and socio-economic development in general. Enke [28] finds a significant association between the Kinship Tightness Index from Murdoc's Ethnographic Atlas and the contemporary level of out-group trust as measured by the WVS. In their most recent paper, Gutmann & Voigt [29] show that communitarian (extended) families have a positive effect on racist and xenophobic attitudes which indicate a lack of out-group trust.

This literature might be placed in a more general context that explores the effect of the historical family on socio-economic outcomes. We could divide this research into two groups, one of them devoted to family extendedness/nuclearity and the other to gender inequality. Todd [14, 30] makes an important contribution, firstly providing a classification of family structures based on such features as extendedness of the family and inheritance type and secondly linking the historical family with economic development. More recently, there has been a wealth of literature emphasizing the advantages of the nuclear family over the extended family. The nuclear family has been shown to be positively associated with economic performance, industrialization, rule of law, educational attainment, social equality, democracy, and the formation of corporations and large business groups [10, 12, 15, 21, 29]. Other researchers oppose these claims and cast doubts on the positive effect of the nuclear family [31, 32].

The second group of studies focuses on the contemporary consequences of historical gender inequality. As a starting point we could consider the seminal work by Hajnal [17], who argued that the European Marriage Pattern–late marriage and a high rate of celibacy–played a positive role in the economic and institutional development of North-Western Europe. Historical gender equality is positively associated with economic development, human capital accumulation, and more intelligent political regulation and reduced corruption, as well as with a higher level of democracy [33–37].

We contribute to the existing literature by disentangling the effects of family extendedness and intra-family hierarchies on out-group trust. Our unique data collection based on historical census microdata allows us to create the most nuanced family indicators. By doing so we provide a valuable addition to the existing data collections [14, 30, 32], which comprise more general family indicators (family complexity, inheritance rule, age at first marriage, lifetime celibacy) readily available from printed historical statistical records or qualitative sources. Therefore, we can answer an additional question that has not been tackled in the previous literature: why are nuclear family arrangements beneficial for out-group trust? Is the effect of small family size and the objective need to cooperate with people beyond the kin group what matters, or is it the lack of strong hierarchical relations preventing trust formation?

## 2.2 Persistence of the family effect

In this section we are going to unmask the "miracle" of how it is possible to find the correlation between family characteristics in the past and social trust more than 100 years later. Bisin & Verdier [38] suggest a transmission mechanism of cultural traits through several generations where family plays a central role. Initially, people may consider particular cultural traits when choosing their marital partner. Once the family unit is established, for some period of time the

parents are the main agents responsible for the socialization of the child. After that, parents are also able to intervene in a child's identity formation by shaping his external environment, for example, by choosing the "right" neighborhood in which to reside, the "right" school in which to learn, the "right" peers, and so on. Bisin & Verdier [38] mention, among others, Catalans, Corsicans, and Irish Catholics in Europe, and Quebecois in Canada, as examples of ethnicities that remained strongly attached to their cultural traits despite being located in political states that neglect their ethnic diversity.

The family pattern itself also has a longstanding tradition. It has been shown that there was a path dependence in cousin-marriage practices for centuries [39], and that cousin-marriage in the past and present have a similar impact on contemporary incidence of democracy and political activity [15]. Thus, it could be the case that the link between historical family structure and trust today is mediated through the contemporary family pattern. Meanwhile, family characteristics in the past might not necessarily be correlated with the same characteristics at present. Some family features that were significant in the past might no longer be indicative. For instance, the potential for a young woman to live neither with her father nor with her husband, working as a servant in another family, was considered to be a move away from patriarchy towards modernization in the 19th century. Nowadays, this indicator does not give any substantial information. Indeed, family forms such as cohabitation before marriage or same-sex marriages could be seen as progressive. Consequently, the level of trust today might be explained by the path dependence between family progressiveness in the past and in the present.

It is possible that out-group trust associated with the pre-industrial family may still exist today, in spite of the discontinuity between past and present family features. This scenario is plausible, as trust and social capital in general display very long-standing characteristics [2, 38]. The persistence of trust may be supported by path-dependence in social institutions triggering trust. The literature offers numerous examples of contemporary institutions logically evolving from historical ones; therefore, it is quite difficult to transplant them from one place to another. For instance, Putnam [2] draws parallels between medieval self-governed communes in Italy and contemporary participation in voluntary organizations, along with the quality of public services. Similarly, La Porta et al. [4] report a path-dependence between the origin of the country's legal system (English, French, German, Scandinavian, or Socialist) and the present-day quality of government. Participatory institutions, good government, and effective law evoke trust by lowering the transaction costs of cooperation between people. Consequently, the path-dependence of trust-supporting institutions could result in outstanding persistence of trust.

## 2.3 Hypotheses

We demonstrate different mechanisms for the effect of family structure on out-group trust, linked to a) family extendedness, and b) internal hierarchy on the basis of gender and seniority.

**Family extendedness and out-group trust.**   There is an influential branch of research that associates family extendedness with less social contact beyond the kin group [10, 12, 14, 22]. It points to the evidence that extended families often represented self-sufficient "production units" and, at the same time, "social cells" that tended to fulfill all the needs of their members, including caring for their children and their elderly, using their own internal capacities [22]. By contrast, nuclear families had to cooperate with unrelated outsiders to satisfy their economic and physical needs and to ensure care for their elderly and children. A greater need for voluntary cooperation with non-kin may have stimulated the education of children in out-

group trust. Through generational value transmission, this educatory thrust could have been passed on from the past until the present.

Extended families served as substitutes for corporations such as monasteries, fraternities or insurance guilds, which emerged as social safety nets against famine, unemployment, and disability in areas dominated by nuclear families [12]. As a corollary, these corporations, extending beyond the kin-group and stimulating intensive out-group contact, were nearly absent in territories dominated by extended families. Likewise, extended families prevented the formation of inclusive political institutions in which people broadly participate in the governing process [15] and the building of large business corporations [21]. In sum, family extendedness that implies a larger number of family members should lead to worse institutional quality and a lower level of out-group trust.

H1: Family extendedness is negatively associated with out-group trust today.

A different logic may complicate this negative association. Family extensions can be both vertical (intergenerational extensions) and horizontal (relatives belonging to the same generation as nuclear family members). Horizontal family extensions involve an enlargement of the relatives' group of the same generation as the nuclear family members and, accordingly, of the same social status based on age. These relatives have to communicate with each other and to reconcile their interests on a daily basis. Putnam argues that interactions between individuals of equal social status foster robust norms of reciprocity and of mutually acceptable behavior. As a result, people are more likely to perceive their counterparts as trustworthy [2]. We believe that Putnam's theory is applicable to household functioning as well. Impartial social norms regulating behavior between equals become even more important in family settings where people are not able to escape from regular interactions with their relatives. The propensity to trust each other emerging within the horizontally extended family may be an important ingredient in trust formation between people who are not related by blood, extending to people of other religions and nationalities or to unknown people generally. This argument is especially plausible in light of Welzel & Delhey's [9] finding that out-group trust is rooted in in-group trust because before people are able to trust everyone, they need to learn to trust someone.

Servants who live in non-kin families constitute an additional type of household extension that was very widespread in Western Europe in past times. The presence of servants is responsible for communication beyond the kin group that also triggers development of a special set of social norms that satisfy all parties and prevent conflicts between people not related by blood ties. This sort of out-group interaction within a household may also nurture out-group trust formation.

Hence, we admit that the "isolative" effect of family extendedness due to higher "self-sufficiency" may be counterbalanced by the "cooperative" effect due to horizontal family extensions and the presence of servants. In cases where these two opposite effects have similar magnitudes and thus cancel each other out, a statistically significant association between family extendedness and out-group trust may be missing. The theoretical mechanisms of these two conflicting effects of family extendedness on out-group trust are presented in Fig 1.

**Family hierarchy and out-group trust.** Power relations within the family, based on gender and seniority, may be detrimental for trust formation. According to Putnam [2], vertical networks cannot sustain trust. On the one hand, sanctions protecting from opportunism are not likely to be imposed upwards. Consequently, as Farrell [40] points out, when a person has full power over another, there is no reason for him to take the interests of this person into account. Thus, the more powerful one of the counteragents, the less he can be trusted. On the other hand, the subordinate person often tries to hide true information in order to avoid exploitation or punishment [2]. As an expected result, the person in power exhibits more distrust. In such a way, a "vicious circle" of mutual distrust becomes an inherent feature of

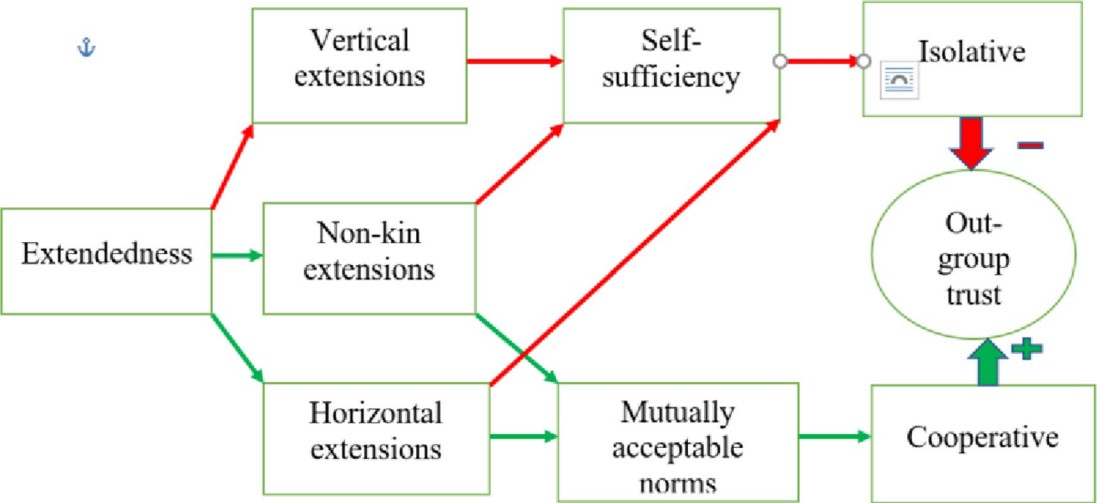

**Fig 1. Theoretical mechanisms of the effect of family extendedness on out-group trust.** Note: This figure summarizes the theoretical mechanisms through which household extendedness may affect out-group trust. The negative path is shown in red. Each type of household extension contributes to its self-sufficiency and low need for out-group contacts. This isolative component may have a negative effect on out-group trust. The positive pathway is shown in green. Horizontal extensions (lateral relatives) and non-kin household extensions (servants) trigger the elaboration of mutually acceptable social norms. This constitutes a cooperative component of household extendedness that positively affects out-group trust.

unequal relations. A good illustration for this is relations between parents and their adolescent children. It has been proposed that parental trust is based on knowledge of children's feelings and concerns, as well as of their daily activities [41]. When children perceive their parents as not trustworthy and are not sincere with them, the parents become even more distrustful.

Lack of trust within the family could lead to a lower level of out-group trust [9]. Bearing in mind that trust is acquired mostly during the formative years [42], we might expect that adults raised in vertically ordered families would have lower trust scores. A large number of people with low out-group trust within a particular territory might lead to less effective formal and informal institutions, sustaining a low trust level. This rationale generates two logically intertwined hypotheses:

H2a. Generational hierarchy within the family is negatively associated with out-group trust today.

H2b. Gender hierarchy within the family is negatively associated with out-group trust today.

## 3. Data, variables, methods

### 3.1 Contemporary data and main dependent variable

As the primary source of contemporary data, we use the LiTS, conducted by EBRD in all post-Communist countries in 2006 and 2010. This survey covers 17 countries of Central, Eastern and South-Eastern Europe and 13 countries from the Commonwealth of Independent States (CIS). We use data from the 2010 round as it contains different questions that allow us to measure out-group trust. The 2010 round also includes, for comparative purposes, 5 Western European countries: France, Germany, Italy, Sweden, and United Kingdom. The total LiTS sample numbers about 38,000 individuals. All microdata are freely available on the official website of the survey [43]. The samples for each country are representative of the adult national residential population.

To measure out-group trust, we replicate the out-group trust index proposed by [6], summing up the trust scores across three items: 1) trust in people you meet for the first time; 2) trust in people of another religion; 3) trust in people of another nationality. In our regressions we also use the standard set of individual socio-demographic controls provided by LiTS (age, gender, education, income level, and type of settlement).

## 3.2 Historical data

The primary sources of information on historical family organization are national historical censuses. First of all, we use microdata coming from the Integrated Public Use Microdata Series, International (IPUMS, International) [44] and the Mosaic project [45]. These data provide information on age, gender, marital status, and relationship to the head of the household for each member of the household, which makes it possible to measure family extendedness and family hierarchies. Most data refer to the 19th century, while some refer as far back as the 17th–18th centuries. Potentially, these data cover 170 subnational units of 8 contemporary countries (7 historical states) covered by the LiTS: Albania, Croatia, France, Hungary, Romania, Slovakia, Sweden, and United Kingdom. However, not all of these subnational units can be matched with localities covered by the LiTS (see the description of the matching procedure below), and so we calculate historical family indicators for only 94 subnational units. This dataset constitutes our core sample.

Additionally, we use published statistics from the aggregated results of historical censuses. These data are available for a maximum of 526 subnational units (289 of which can be matched with the LiTS 2010 sample) in 26 contemporary countries (14 historical states) covered by the LiTS, including Belarus, the Czech Republic, France, Germany, Hungary, Italy, Russia, Serbia, and Ukraine. However, the aggregated census data allow us to construct only a limited list of family indicators–mean household size, percentage of single households, percentage of households with servants, and percentage of never-married women aged 20–29 –and do not allow us to differentiate between family extendedness and generational hierarchy. Therefore, these data are not suitable for testing Hypotheses H1 and H2a, while Hypothesis H2b is partially testable (by using the percentage of never-married women aged 20–29). These data constitute our extended sample, used for the robustness check when testing Hypothesis H2b and also for descriptive purposes (e.g., to trace the famous Hajnal [17] line, that runs from Trieste (Italy) to Sankt Petersburg (Russia) and divides Europe into two areas characterized by different marriage and household arrangements).

Table 1 provides the geographical composition of the core and extended samples, while Table S1.1 in the S1 File presents a detailed description of the sources of historical family data.

(A detailed description of each variable, as well as the number of observations and its assignment to the composite indices, can be found in Table 2. Table S1.2 in the S1 File shows the availability of historical family variables for different countries and time periods).

All the historical family indicators that we collected fall into five large groups.

*Group 1*: *Family extendedness*. This group contains different indicators measuring mean household size at the regional level. These indicators may include all the household types and count all the people living in the dwelling or exclude (a) certain types of households (e.g., one-person households), (b) certain categories of people (e.g., persons who live in the dwelling but are not related by blood ties, such as servants or lodgers).

*Group 2*: *Household composition*. This group includes two indicators reflecting the composition of households: 1) the proportion of children and adults in a household and 2) the proportion of households that have servants.

**Table 1. The geographical composition and the number of subnational units in the core and extended samples.**

|  |  | Extended sample | Core sample |
|---|---|---|---|
|  | Contemporary countries | Historical regions/subnational units | |
|  | France | 56 | 14 |
|  | Germany | 33 | |
|  | Great Britain | 41 | 41 |
|  | Italy | 16 | |
|  | Sweden | 18 | 18 |
| Total Western Europe | 5 | 164 | 73 |
|  | Albania | 3 | 3 |
|  | Armenia | 1 | |
|  | Azerbaijan | 2 | |
|  | Belarus | 4 | |
|  | Croatia | 3 | 1 |
|  | Czech Republic | 3 | |
|  | Estonia | 1 | |
|  | Georgia | 3 | |
|  | Hungary | 4 | 4 |
|  | Kazakhstan | 5 | |
|  | Kyrgyzstan | 2 | |
|  | Latvia | 2 | |
|  | Lithuania | 3 | |
|  | Moldova | 1 | |
|  | Poland | 18 | |
|  | Romania | 5 | 5 |
|  | Russia | 29 | |
|  | Serbia | 16 | |
|  | Slovakia | 8 | 8 |
|  | Slovenia | 2 | |
|  | Ukraine | 11 | |
|  | Uzbekistan | 2 | |
| Total Eastern Europe | 22 | 128 | 21 |
| TOTAL | 26 | 292 | 94 |

*Group 3*: *Horizontal extensions*. This group accounts for horizontal (lateral) household extensions. Lateral relatives are family members of the same generation as the nuclear family members (e.g., brothers, sisters, aunts, uncles, cousins, nieces, and nephews). Households with horizontal extensions are two-generational households, like nuclear families. We calculate the proportions of households with horizontal extensions in a historical region both including and excluding vertical extensions.

*Group 4*: *Vertical extensions*. This group contains indicators related to households comprising more than two generations. We assign this group a generational hierarchy index (see more about index construction in Section 3.3) and the percentage of families with vertical extensions (adult or married children, children-in-law, grandchildren, grandparents, grandparents- and grandchildren-in-law, etc.).

*Group 5*: *Gender hierarchy*. We include in this group all variables associated with gender (in)equality that are part of the gender hierarchy index (e.g., percentage of female household heads in the region, percentage of wives who are older than their husbands. See more about the gender hierarchy index in Section 3.3).

**Table 2. Historical family indicators and their description.**

| Indicator | Description | Max. N of subnational units for which this indicator is available | Form of inclusion in regression model |
|---|---|---|---|
| **Group 1: Family extendedness (number of relatives)** | | | |
| Mean HH | **Mean household size.** The household comprises all persons who live in the same dwelling. Besides the kin group, servants, lodgers and other non-relatives might belong to the household. This measure counts all the persons living in the household. | 178 | Independent explanatory variable |
| MeanHH no one person HH (Ext)* | **Mean household size with no one-person households**. We exclude from this measure one-person households in order to distinguish the situation 1) when a large number of extended families is comes together with a large number of one-person households; and the situation 2) when nuclear families are prevalent but the number of one-person households is very small. | 447 | Independent explanatory variable |
| Mean HH no children | **Mean household size excluding children aged 0–14.** This measure is needed to capture the self-sufficiency of the household. Therefore, we single out adult household members from children who cannot be full-value workers. | 170 | Independent explanatory variable |
| Mean kin group size no one person HH | **Mean kin group size**. Household members who are not related by blood ties are excluded from this measure. One-person households are excluded. | 170 | Independent explanatory variable |
| one person HH (Ext) | **One-person households.** The proportion of households comprising one person. | 447 | Independent explanatory variable |
| **Household composition** | | | |
| Children/adults | **The number of children aged 0–14 divided by the number of adults**. This variable measures the self-sufficiency of the household. The larger this ratio the less self-sufficient the household, as additional adults are needed to provide food and child care for the large number of children. This variable can be also used as a control for the number of children within the household. | 170 | Independent explanatory variable, control variable |
| Servants (Ext) | **Percentage of households having servants.** | 266 | Independent explanatory variable |
| **Horizontal family extensions** | | | |
| Lateral | **Lateral household extension.** Lateral relatives are family members of the same generation as the nuclear family members: brothers, sisters, aunts, uncles, cousins, nieces, nephews. We measure the percentage of households that have lateral extensions. | 170 | Independent explanatory variable |
| Lateral no vertical | **Lateral relatives as the only household extension.** We measure the percentage of households which have lateral extensions but do not have vertical linear extensions as measured by the variable "*vertical extensions*". This is a measure which enables us to distinguish households with horizontal extensions from multi-generational households. | 170 | Independent explanatory variable |
| **Vertical family extensions** | | | |
| Vertical hh, all | **All possible vertical household extensions.** The percentage of households that have any type of vertical extension: parents, parents-in-law, adult (aged 20+) or married children, children-in-law, grandchildren, great-grandchildren. | 170 | Generational hierarchy index |
| Vertical hh, father head | **Vertically extended households headed by father.** The percentage of households headed by the oldest man in the household that have lineal (vertical) family extensions associated with patriarchal family. We count adult children (20+), married sons, grandchildren and great grandchildren. We exclude grandparents from this measure, because if they are mentioned in the census list as grandparents it means that they live in a household headed by their child. This case contradicts the patriarchal family rule where the household head should be the oldest man in the family. Likewise, married daughters are not counted because according to the patriarchy principle a woman should move to live with her husband upon marriage. | 170 | Independent explanatory variable, part of the generational hierarchy index |

(*Continued*)

**Table 2.** (Continued)

| Indicator | Description | Max. N of subnational units for which this indicator is available | Form of inclusion in regression model |
|---|---|---|---|
| Vertical hh, son head | **Vertically extended households headed by son.** The percentage of vertically extended households headed by the son and comprising his father. If the man of the older generation is listed as "father of the household head" it means that the household is headed by his son. | 170 | Generational hierarchy index |
| Prevalence of vertical households headed by son | **Prevalence of vertically extended households headed by son over households headed by father.** Vertical hh, son head/ Vertical hh, father head | 170 | Generational hierarchy index |
| **Gender hierarchy** | | | |
| women20_29 (Ext) | **Percentage of single women in the age group 20–29.** This is an alternative measure of age at first marriage and it is a proxy for women's emancipation. The older the woman when she gets married the more independent she might be from her husband (Gruber, Szołtysek, 2016). | 449 | Gender hierarchy index |
| female hh heads (G)** | **Percentage of female household heads among all the adult (20+) household heads.** Rescaled variable so that higher score means fewer female household heads. | 224 | |
| young brides (G) | **Percentage of married women aged 15–19**. This is also a proxy for age at first marriage. | 224 | |
| wives older (G) | **Percentage of wives who are older than their husbands**. Rescaled variable so that higher score means fewer old wives. | 224 | |
| female non kin (G) | **Proportion of young women living as non-kin.** Proportion of women aged 20–34 years who live as non-kin, usually as lodgers or servants. These women are not controlled by their relatives or by their husband's relatives. Rescaled variable so that higher score means less women living as non–kin. | 224 | |

Note: *Variables available for the extended sample have the sign (Ext).

**Variables borrowed from Gruber and Szołtysek 's (2016) patriarchy index are marked with (G). All variables marked with (G) are rescaled by the authors of the patriarchy index to a range from 0–10 so that higher values mean more gender inequality. In this table is given the maximum number of observations available for each indicator. We matched only 94 regions for the core sample and 292 regions for the extended sample with the LiTS dataset.

## 3.3 Main explanatory variables

Family extendedness is captured by different measures of mean household size listed in Table 2. Family extendedness indicates horizontal family extensions in the model when it is controlled for gender and generational hierarchies. Alternatively, we construct a special measure that includes only horizontal (lateral) family extensions (*lateral no vertical*).

Our indices of generational and gender hierarchies are constructed by drawing the first principal component from several family structure indicators (see S2 File and Table S2.1 for the Principal Components Analysis results). The generational hierarchy index captures vertical family extensions and the role distribution between father and son in vertically extended households. We assume that the younger generations are more influenced by the older generations when they live together in an extended household compared to living in a nuclear family. Therefore, one component of the generational hierarchy index is the percentage of households in a region comprising any sort of vertical extension, like parents, parents-in-law, adult or married children, children-in-law and (great) grandchildren (*vertical hh, all*). The second component accounts for the distribution of power between father and son in vertically extended households. We calculate the percentage of households where the son is the household head while his father is listed as "parent" (*vertical hh, son head*). Then we calculate the proportion of vertically extended households headed by the father and comprising adult

children, married sons and grandchildren (*vertical hh*, *father head*). The ratio of *vertical hh*, *son head* to *vertical hh*, *father head* shows which type of multigenerational household prevails in a given region. It is the second component of the generational hierarchy index (prevalence of *vertical hh headed by son*). To combine these two items into an index we draw the first principal component. The first item (*vertical hh*, *all*) is positively associated with generational hierarchy while the second item (*prevalence of vertical hh headed by son*) is negatively linked with this concept.

The gender hierarchy index combines all items associated with gender inequality and the obedient status of women. First, we consider the proportion of female household heads among all the adult household heads, which would indicate the independent status of women. The next two items are proxies for age at first marriage for women. The idea behind these items is that an adolescent wife is more likely to obey her husband and his relatives than an adult woman who has had time to accumulate some wealth before marriage. We use the proportion of married women aged 15–19 and the proportion of never-married women aged 20–29. Following the similar logic that older wives are less obedient we include an additional item that measures the percentage of wives who are older than their husbands. We also add to our index the proportion of young women aged 20–34 who live in non-kin families as lodgers or servants. This is a measure of women's emancipation, as in a patriarchal society a young woman would live either with her father or with her husband. Similarly, to the generational hierarchy index we draw the first principal component from all these items. Four gender hierarchy items *(% of female household heads, % of married women 15–19, % of wives who are older than their husbands, % of young women 20–34 who live in non-kin families)* drawn from Gruber and Szołtysek's [46] patriarchy index were originally rescaled so that higher scores mean more gender inequality. These items positively affect the factor score of gender hierarchy. The percentage of never-married women aged 20–29, which is the proxy for higher age at first marriage, negatively affects the factor score.

The correlation between gender hierarchy and generational hierarchy indices is r = -0.6***, but it vanishes when we control for the country fixed effects (r = -0.02) (This correlation refers to a more comprehensive sample of 170 regions. For the core sample (94 regions matched with the LiTS) the correlations are -0.44*** without country FE and -0.03 with country FE). A possible explanation might be that gender hierarchy is revealed to be more sensitive to the modernization and industrialization processes of the 19th century than the more rigid generational hierarchy rooted in clan society. Therefore, a relative gender equality, driven to a large extent by societal factors, could temporally coexist with traditional family arrangements such as the co-residence of adult children with their parents or households headed by the oldest man. The relationship between gender equality and generational hierarchy deserves special study; nevertheless, they are clearly two different indicators that should be studied separately.

## 3.4 Matching of historical and contemporary data

The essential problem with linking regional historical indicators with contemporary variables is that the borders of countries, and especially the borders of regions within countries, have been changing over time. To solve this problem, we use the shapefile of historical regions of Europe for the year 1900, available from the official Mosaic project website, and create the map using R software. As the next step, we compile a data file with geographical coordinates for each contemporary locality from the LiTS (the lowest level of aggregation available in the LiTS) and place these localities as points on a historical map of Europe. Finally, we assign the set of our historical indicators to each contemporary locality.

## 3.5 Econometric modeling

We estimate the following equation:

$$\text{Trust}_{irc} = \beta_0 + \beta_1 * \text{Generational hierarchy}_{rc} + \beta_2 * \text{Gender hierarchy}_{rc} + \beta_3 * \text{Family extendedness}_{rc}$$
$$+ \text{Individual controls}_{irc} + \text{Regional controls}_{rc} + \alpha_c + u_{irc} \tag{1}$$

where $i$ refers to individuals, $r$ to historical regions, $c$ to historical countries, $\alpha_c$ accounts for the countries fixed effects (FE) and $u_{irc}$ is the conventional error term. We include historical country FE instead of FE for the contemporary countries to better control for omitted variables which may affect both historical family structures and out-group trust. Obviously, contemporary societal characteristics do not affect historical family structures but rather may be influenced by them. Historical country FE account for various unobserved country-level characteristics, including historical institutional quality and state capacity and the different pace of the Industrial Revolution, plus the year of the observation of historical variables. Eq 1 is estimated using OLS taking into account possible correlation of errors within historical regions (OLS-CRSE).

We use different measures of family extendedness at the subnational (regional) level listed in Table 2: (a) all people in a dwelling/number of households *(Mean HH)*; (b) all people in a dwelling/number of households excluding one-person households *(Mean HH no one person HH)*; (c) all people related by blood ties living in a dwelling/number of households excluding one-person households *(Mean kin group size no one person HH)*; (d) all adults in a dwelling/ number of households *(Mean HH no children)*; (e) percentage of households which have lateral and no vertical extensions *(Lateral no vertical)*. If one-person households or servants are excluded from the measure, we include them in the model as separate predictors. We are mainly interested in adult household members as the self-sufficiency of an extended household (Hypothesis H1) relies on adult labor. When we include indicators that do not explicitly exclude children in our regressions, we add children/adults ratio among the controls.

Individual-level controls include age, gender, higher education, income level, and urban/ rural settlement.

In our core specification at the level of historical regions, we use population density as a measure of urbanization because we expect that it may have an impact on family structure as well as on out-group trust. An urban environment may be associated with smaller families as there was no need for a large number of agricultural laborers [47]. Simultaneously, people from different regions moved to the urban centers and consequently, their trust in out-groups might be higher than in rural areas.

Further, we control for geo-climatic conditions. On the one hand, they are an important determinant of economic development in general [48] and its main constituents, such as out-group trust, and on the other hand, they may have shaped the family structure [49]. We use the Cool Water Index, proposed by Welzel et al. [49], which combines temperature and precipitation components to measure geo-climatic conditions favorable for cultural and institutional modernization. Simply put, the index measures the combination of cool summers with temporarily frosty winters but with continuous rainfall throughout all seasons.

As a robustness check we add additional regional-level controls that may be correlated with the historical family indicators as well as with modernization and its components. We control for the caloric suitability of land for agriculture [50], because agriculture influences people's values and attitudes [51] and creates favorable conditions for the extended family [28]. A further control variable is ruggedness of the terrain [52], indicating remoteness of the region, which may affect psychology and family structure. It is well-documented in the literature that remote mountainous territories are conducive to an extended family because of the patriarchal

way of living and high insecurity due to lesser penetration of the state [53, 54]. We also control in our models for proximity to waterways. These indicators approximate market integration of the territory, which could jointly stimulate the nuclear family and trust towards out-groups. Welzel et al. [49] provide evidence that territories well-suited for hunting and gathering adopted agriculture several thousand years later than the first adopters in the Middle East, which ultimately created favorable conditions for the Industrial Revolution and left an imprint on contemporary social values and institutions. Family structure may also be affected by a long tradition of hunting and gathering, as foraging societies featured nuclear families [28] due to low demand for labor and food scarcity. Therefore, we control for the prevalence of land suitability for hunting and gathering over agriculture [55]. Finally, we include exposure to the medieval Catholic Church as a control variable, as it has been shown to affect kinship systems as well as social trust [15]. All these control variables were included in the model one by one in order to avoid multicollinearity. The sources of all regional level controls are summarized in S1 File and Table S1.3.

As a further robustness check we run a regression on an extended sample where we have a larger number of observations but a limited number of family indicators. Unfortunately, we are not able to disentangle family extendedness and power hierarchy in the extended sample. Thus, in this case, we include in our model only the percentage of never-married women aged 20–29. In other respects, we use the same specification as shown in Eq (1).

One of the limitations of our study is that LiTS is not representative at the subnational level. If this leads to non-classical measurement errors in the dependent variable (out-group trust), this may bias our estimates. However, since the two-stage quasi-random sampling procedure used in LiTS involves random sampling of households within a PSU (which is usually an electoral territorial unit), these measurement errors should not be systematic and correlated with any factors, and thus should not cause any bias. In order to be completely on the safe side, we also provide a check to ensure that LiTS regional scores on our dependent variable do not differ considerably from the scores on the same variable coming from representative sources. For this purpose, we took the "Values in the Russian Regions" survey [56] conducted in January–February 2019, which is representative of 60 (in 2019, out of 85) Russian regions and has exactly the same questions that we use to construct the out-group trust index in LiTS. We aggregated the data at the level of the Russian federal districts in order to obtain the same administrative units used in LiTS. The S1 Fig shows quite a strong correlation (r = 0.83***) for out-group trust coming from representative ("Values. . ." ) and unrepresentative (LiTS) sources. This indicates minimal measurement errors in trust variable due to unrepresentativeness of LiTS at the subnational level.

## 4. Main findings

### 4.1 Descriptive analysis

Figs 2 and 3, which are constructed using the extended sample, illustrate Hajnal's [17] idea of the existence of a specific West European marriage pattern and household formation system. While Hajnal's analysis was based only on a few case studies, we use data derived from national historical censuses covering almost all of Europe. Our maps show that to the east of Hajnal's line, households were larger and the share of never-married women aged 20–29 was lower, reflecting lower age at first marriage. However, to the west of Hajnal's line, the picture looks more heterogeneous, contradicting the idea of a uniform Western European family pattern.

Figs 4 and 5 present the level of vertical family extensions (% of vertically extended households headed by father and the generational hierarchy index) and horizontal family extensions across 170 regions for which individual data are available (The same figures constructed for

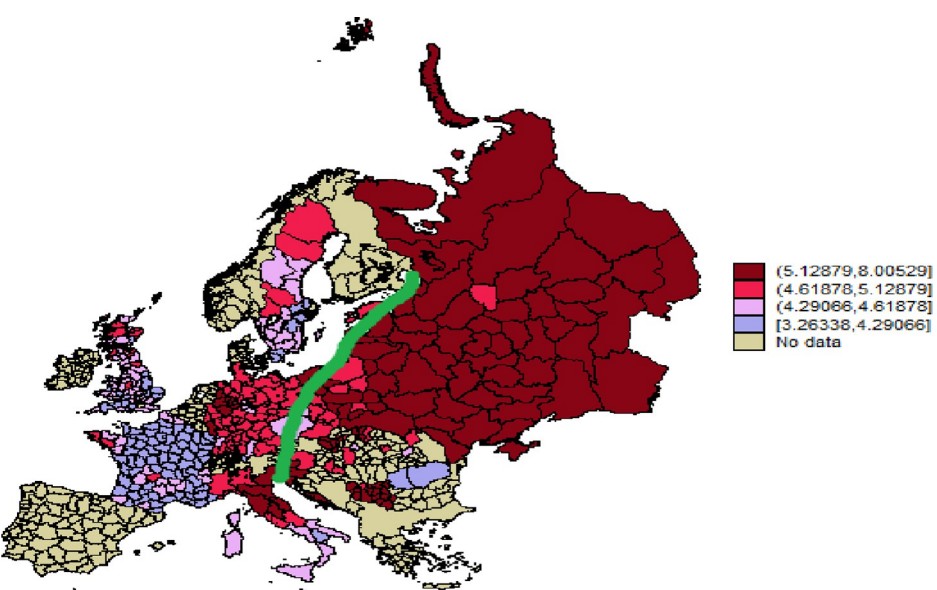

**Fig 2. Mean household size in Europe divided by Hajnal's line.** Republished from https://censusmosaic.demog. berkeley.edu under a CC BY license, with permission from Mag. Dr. Siegfried Gruber, original copyright 2013. Note: On this map we plot mean household size excluding one person households for the historical Europe. The green line is the Hajnal's line that runs from Trieste (Italy) to Sankt-Petersburg (Russia) and singles out the West European marriage and household formation pattern. The data shown on the map were collected by the authors of the present article. The shape file with regional boundaries for Europe 1900 was republished from https://censusmosaic.demog. berkeley.edu.

our core sample are available in the Appendix (see S2 and S3 Figs). They do not considerably differ from Figs 4 and 5). Taking a rapid glance at the figures we can see that our regional observations are highly clustered at the country level. This may partly reflect the country-level situational conditions in the past as well as more deeply-rooted differences. In Fig 4 the y axis represents the percentage of vertically extended households headed by the father. These are patriarchal households where older generations are in a dominant position over younger ones. According to our theory, such power inequality may be harmful for trust. The x axis reflects the percentage of households having horizontal extensions, which are more favorable for trust. The reference lines shown in red are placed at the mean scores for horizontal and vertical extensions. The scores on the x and y axes are deviations from the mean. According to Fig 4, Albanian regions are the absolute champions in vertically extended households, which is to be expected as the Balkans are known for their large multigenerational families, "zadruga" [57]. The proportion of vertically extended households headed by the father in France and Scotland is considerably lower than in Albania but higher than in the other countries in our sample (See S4 Fig in the Appendix for the incidence of lateral and vertical family extensions across European countries excluding Albania).

The situation is very similar in case of horizontal family extensions. Albania is far beyond the average while France and Scotland have a higher level of horizontal extensions than the other countries. Sweden, England and Wales, together with Wallachia (Romania), score quite low on both family indicators. The lack of extended families in Wallachia (Romania) seems surprising at first glance. In fact, in Romania, like southern Italy and many other countries of the Eastern Europe, the nuclear family had a different nature. It emerged not as a consequence of unigeniture where only one offspring inherits the farm (Great Britain, Sweden, Scotland),

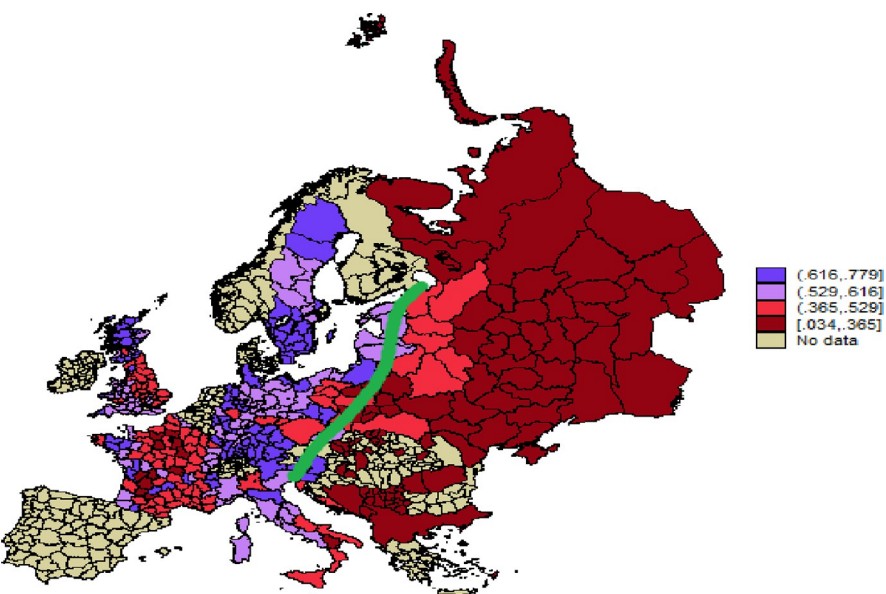

**Fig 3. The percentage of never-married women aged 20–29 (proxy for age at first marriage) in Europe divided by Hajnal's line.** Republished from https://censusmosaic.demog.berkeley.edu under a CC BY license, with permission from Mag. Dr. Siegfried Gruber, original copyright 2013. Note: On this map we plot the percentage of never married women in the age group 20–29 for the historical Europe. This indicator approximates age at first marriage. The green line is the Hajnal's line that runs from Trieste (Italy) to Sankt-Petersburg (Russia) and singles out the West European marriage and household formation pattern. The data shown on the map were collected by the authors of the present article. The shape file with regional boundaries for Europe 1900 was republished from https://censusmosaic.demog.berkeley.edu.

but as the result of partible male inheritance where the parental farm is equally divided among all male heirs. Therefore, in the latter case, the households might be formally nuclear, but they could be not fully autonomous, having a common border with the farms of relatives.

When we include the generational hierarchy index instead of the proportion of vertically extended households headed by the father, the picture becomes slightly different (Fig 5). The regions of England and Wales, Scotland, Albania and (partly) France have a level of generational hierarchy above the average. Albania does not considerably outperform all the other countries in generational hierarchy index capturing all types of vertical extensions (See S5 Fig in the Appendix for the incidence of generational hierarchy and lateral extensions across European countries excluding Albania).This finding is presumably associated with high population pressure in rapidly-industrializing countries like England, Scotland and France. In our econometric analysis we account for these country level differences in industrialization pathways in order to single out the association between the longstanding historical family arrangements and the today's level of out-group trust.

The correlations between family extendedness and intra-family power hierarchy measures are presented in Table 3. Family extendedness is only moderately correlated with generational hierarchy (from 0.30 to 0.56 in the extended sample and from 0.2 to 0.47 in the core sample), which illustrates the idea that not all extended households have hierarchical features. Moreover, mean household size, with or without children, is not correlated with gender hierarchy, which casts doubts on Hajnal's argument that women's obedience to men usually comes together with an extended household type.

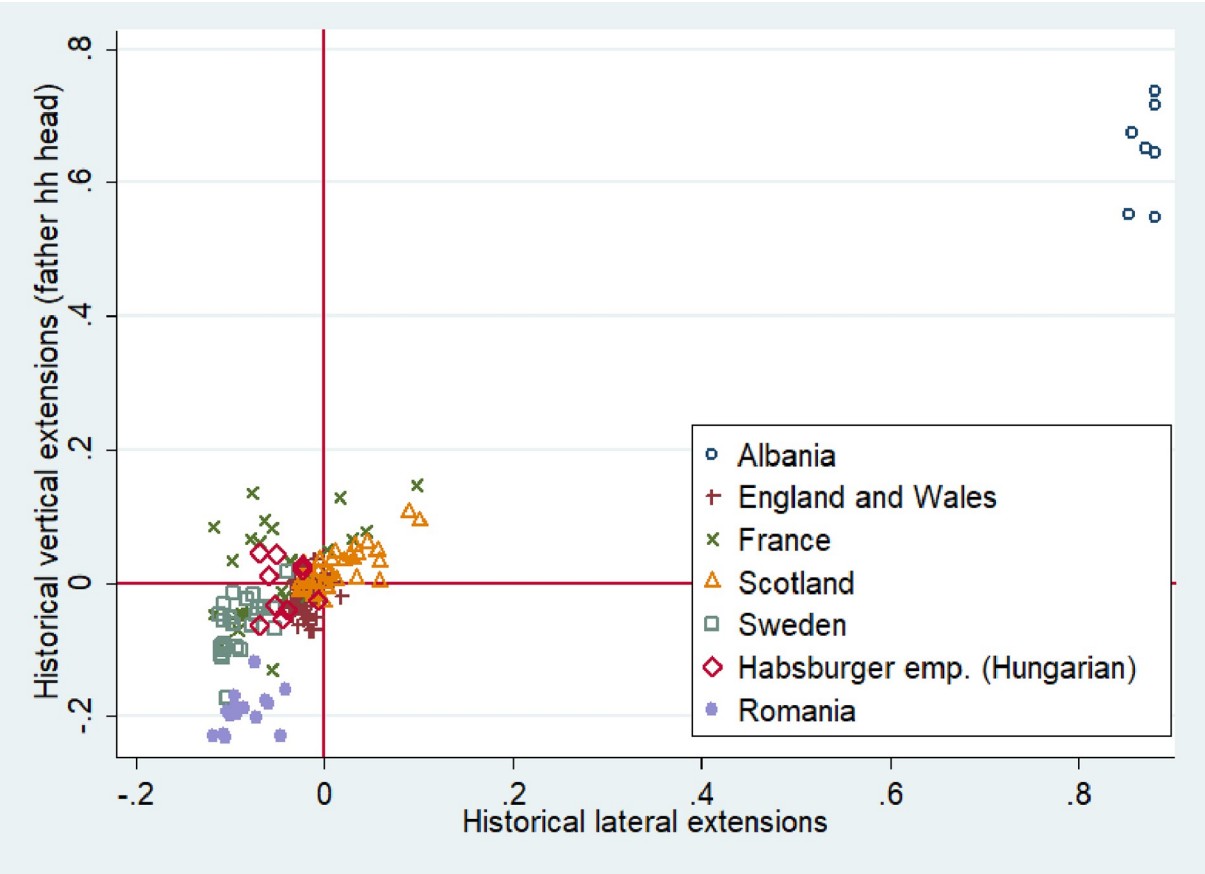

**Fig 4. Incidence of vertical and lateral extensions across 170 regions of Western and Eastern Europe.** Note: Both variables were rescaled so that 0 is the mean value. On the x-axis is shown the percentage of historical households that have lateral (horizontal) extensions. On the y-axis is shown the percentage of historical vertically extended households headed by the oldest man in the household. This Figure is based on the maximum number of regions for which individual census data were available (170 regions).

In addition, we link family extendedness and family hierarchies with our outcome variable of interest, i.e., the contemporary level of out-group trust. To illustrate the persistence effect between the past and the present we apply a more reliable approach focusing on intra-country variation in historical family structure. We consider the fact that different countries were observed at different points in time while they were undergoing different stages of modernization and demographic transition. Country fixed effects account for these differences, eliminating the "between" effect which may reflect temporal trends in the past instead of longstanding traditions. A partial correlation between generational hierarchy and the contemporary level of out-group trust is presented in Fig 6 (r = -0.32***). Partial correlations of gender hierarchy and family extendedness with contemporary out-group trust levels are shown to be insignificant.

## 4.2 Econometric analysis

We do not find that family extendedness is linked to out-group trust (see Table 4). None of the five family extendedness measures yield significant coefficients, and so Hypothesis H1 is rejected. According to our theoretical reasoning, we do not observe the hypothesized negative correlation as the "cooperative" component associated with horizontal family extensions and

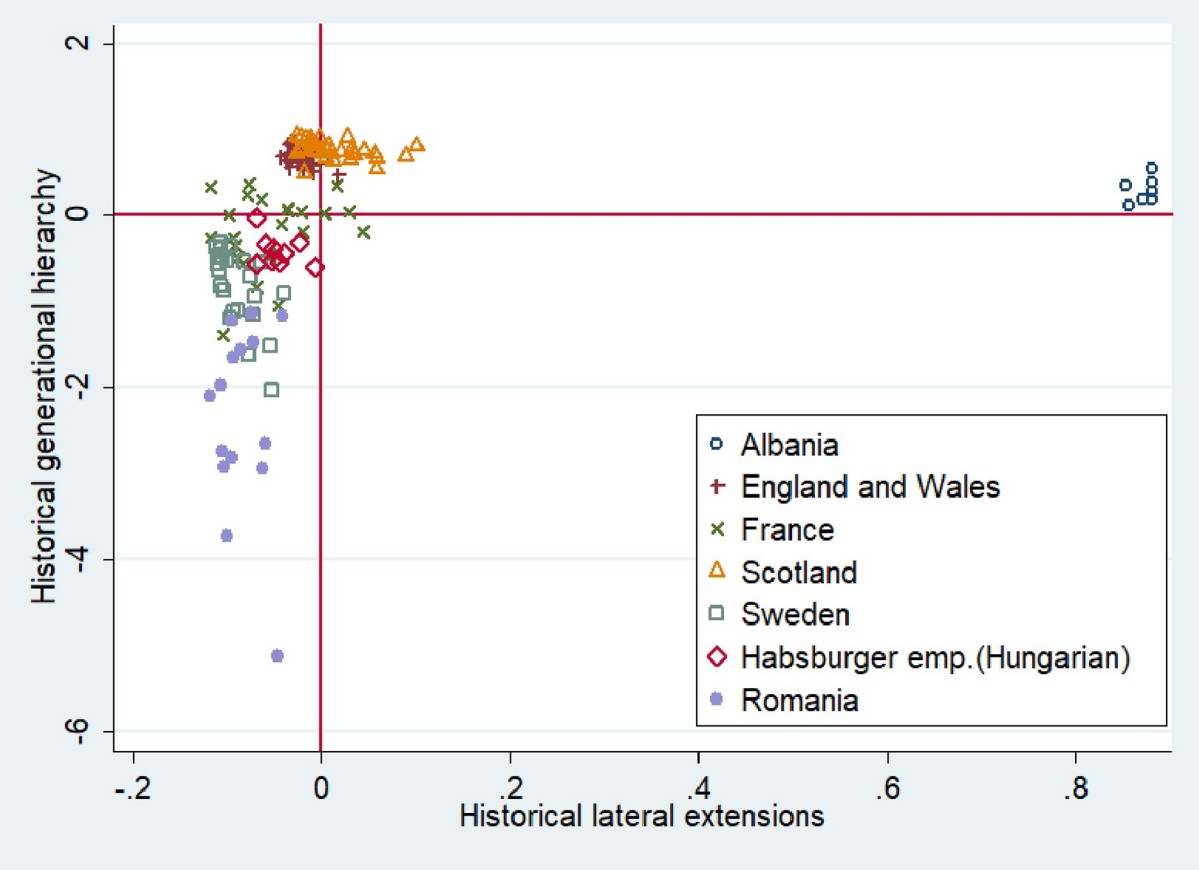

**Fig 5. Incidence of generational hierarchy and lateral extensions across 170 regions of Western and Eastern Europe.** Note: Both variables were rescaled so that 0 is the mean value. On the x-axis is shown the percentage of historical households that have lateral (horizontal) extensions. On the y-axis is shown the generational hierarchy index scores.

non-relatives counterbalances the "isolative" component rooted in the self-sufficiency of the extended family.

At the same time, we establish that generational hierarchy is detrimental to out-group trust, confirming Hypothesis H2a. This result holds in all estimated specifications. The size of the effect is quite substantial. Increasing the generational hierarchy index by one standard deviation decreases out-group trust by about 30% of its standard deviation. The effect of

**Table 3. Correlations between family extendedness and within family hierarchy measures.**

|  | Extended sample | | Core sample | |
|---|---|---|---|---|
|  | Generational hierarchy | Gender hierarchy | Generational hierarchy | Gender hierarchy |
| Mean household size | 0.302*** | 0.006 | 0.198* | 0.160 |
| Mean household size without children | 0.561*** | -0.066 | 0.473*** | 0.144 |
| Lateral (horizontal) extensions | 0.404*** | 0.592*** | 0.252** | 0.262** |
| N of subnational units | 170 | | 94 | |

Note: *p < .05

**p < .01

***p < .001

generational hierarchy remains statistically significant after controlling for various geographical and historical indicators (Table 5). The magnitude of the effects remains in the same range.

As a robustness check, we include two components of the generational hierarchy index–*vertical hh*, *all* and *prevalence of vertical hh headed by son*–separately in the basic specification of our model (Table 6). We find that the proportion of vertically extended households has the predicted negative effect on out-group trust, while the effect of *prevalence of vertical households headed by son* positively affects out-group trust. Comparing the standardized beta coefficients suggests that the effect of the percentage of vertically extended households is considerably stronger (-0.399***) than the effect of *prevalence of households headed by son* (0.09*).

Our findings are best comparable with the previous literature that builds on extended family indicators incorporating co-residence of adult children with their parents (an element of our generational hierarchy index). With our new empirical data, we validate the existing knowledge that the nuclear family, which by definition lacks generational hierarchy, is beneficial for out-group trust formation [10, 15, 28, 29]. Additionally, we provide new evidence that this positive effect is due to the lack of generational hierarchy within the nuclear family. By contrast, in an extended family the dominance of older generations over younger ones prevents trust formation.

We do not find a significant correlation between gender hierarchy and out-group trust (Tables 4 and 5). The result remains robust to the different model specifications on the core sample. We also use an alternative measure of the gender hierarchy index, excluding the proportion of female-headed households, which could capture other factors than women's obedience, for example the incidence of wars. The results remain robust to this procedure (see S3 File and Table S3.1). Hence, we must admit that Hypothesis H2b is not confirmed. However, the proportion of never-married women in the 20–29 age group (one component of the gender hierarchy index available for a larger number of countries) has an expected, positive and statistically significant impact on out-group trust in the extended sample in a specification without controls (see S3 File and Table S3.2, Column 1). This suggests that the effect of gender hierarchy on out-group trust should be investigated further in future studies covering a larger number of countries. This is particularly important when considering that the literature does imply a negative effect of gender hierarchy on the whole modernization complex, including out-group trust [17, 33–37].

The results for gender hierarchy obtained using the core sample may also have a substantive explanation. It could be the case that the effect of generational hierarchy is stronger than that of gender hierarchy. First, the authority of older over younger generations multiplies the dependent status of a young woman because she has to obey not only her husband but also the household head and the oldest woman in the house. Thus, generational hierarchy amplifies the effect of gender hierarchy. Second, generational hierarchy might be costlier to support because it often implies obedience of a young and able-bodied man to an old physically or mentally weak parent. In this situation, adult children keep their dependent position, even though their contribution to the common family wealth is much greater than that of the household head. It is evident that considerable power resources are needed to maintain such an "unfairness". A gender hierarchy could be more easily justified in a pre-industrial society as women have less physical strength and their contribution to agricultural work might be objectively somewhat smaller. We suggest that the generational hierarchy's harmful effect on social trust might be greater requires more power concentration.

The reported results are validated using a number of further robustness checks. Firstly, our findings remain robust to the inclusion of various measures of our main concepts (Table 4). Horizontal family extensions are modelled in two different ways: (a) the mean household size (different versions) controlled for generational family hierarchy (see Columns 5, 6, 8, and 9 of

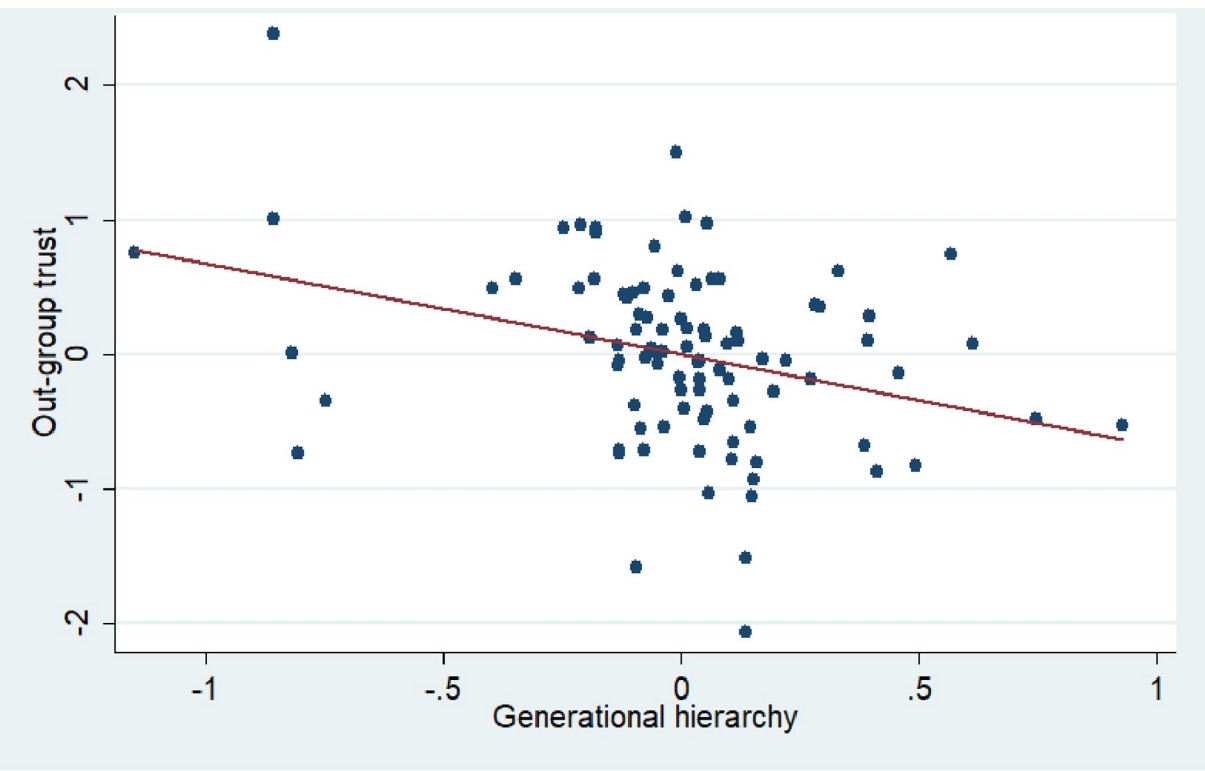

**Fig 6. Partial correlation between generational hierarchy index and contemporary out-group trust.** Note: On this figure we plot residuals from the regressions where generational hierarchy index and out-group trust are controlled for the historical country fixed effects. We eliminate the effect of historical countries because they are very different in terms of the year of observation, industrialization, inheritance type and the stage of demographic transition.

Table 4) or (b) a special indicator of percentage of households having lateral (horizontal) and no vertical extensions (see Column 7 of Table 4). In both cases horizontal extensions are revealed to not be detrimental to trust.

Secondly, instead of the generational hierarchy index, we include in our model the percentage of vertically extended households. The results follow a similar pattern, but the significance level is lower, as this measure does not take into account many nuances captured by the generational hierarchy index (These results are available upon request).

## 5. limitations

One of the limitations of the present study is that it was based on eight contemporary Western and Eastern European countries for which historical data were available. Thus, additional studies are needed to test whether the obtained results can be generalized. A second is that the LiTS is not representative at the subnational level. While we provide theoretical arguments that this fact should not cause any bias in our econometric estimates and also provide a special check, we admit that this limitation may have some effect on our findings.

The third limitation is that our data do not allow us to establish causality between historical family type and contemporary out-group trust. It might be the case that the authoritarian family type was a reaction to a low trust level in society. Alternatively, out-group trust level and particular family arrangements could be associated with a common factor that we do not control for in our regression. To cope with this problem, we included in our models all possible

**Table 4. OLS regression of contemporary out-group trust on historical family indicators.**

| DV: out-group trust (LiTS, 2010) | (1) | (2) | (3) | (4) | (5) | (6) | (7) | (8) | (9) |
|---|---|---|---|---|---|---|---|---|---|
| Generational hierarchy | -0.703*** | | -0.714*** | -0.730*** | -0.742*** | -0.738*** | -0.773*** | -0.720*** | -0.729*** |
| | (0.178) | | (0.184) | (0.188) | (0.204) | (0.207) | (0.241) | (0.190) | (0.201) |
| Gender hierarchy | | -0.00571 | 0.237 | 0.0814 | -0.00488 | 0.244 | 0.429 | 0.00525 | 0.339 |
| | | (0.118) | (0.226) | (0.278) | (0.264) | (0.329) | (0.319) | (0.298) | (0.381) |
| Mean HH | | | | | -0.229 | | | | |
| | | | | | (0.189) | | | | |
| Mean HH no children | | | | | | | | -0.152 | |
| | | | | | | | | (0.246) | |
| Mean HH no one person HH | | | | | | 0.107 | | | |
| | | | | | | (0.214) | | | |
| Mean kin group size no one person HH | | | | | | | | | 0.160 |
| | | | | | | | | | (0.224) |
| One person HH | | | | | | 3.397 | | | 3.570 |
| | | | | | | (2.346) | | | (2.290) |
| Children/adults | | | | | | -1.559 | -1.969 | | -1.849 |
| | | | | | | (1.564) | (1.471) | | (1.563) |
| Lateral no vertical | | | | | | | 2.215 | | |
| | | | | | | | (2.874) | | |
| Servants | | | | | | | | | 1.192 |
| | | | | | | | | | (1.843) |
| Population density | | | | -8.18e-06 | -6.26e-05 | -0.000138 | -4.09e-06 | -3.50e-05 | -0.000131 |
| | | | | (4.47e-05) | (5.96e-05) | (0.000102) | (4.41e-05) | (6.43e-05) | (0.000103) |
| CWI | | | | -1.345 | -1.255 | -1.191 | -1.069 | -1.332 | -1.229 |
| | | | | (1.119) | (1.025) | (1.028) | (1.031) | (1.081) | (1.050) |
| Individual level controls | YES | | | | | | | | |
| Historical country FE | YES | | | | | | | | |
| Observations | 4,214 | 5,839 | 4,204 | 4,042 | 4,042 | 4,042 | 3,801 | 4,042 | 4,042 |
| R-squared | 0.221 | 0.182 | 0.222 | 0.234 | 0.235 | 0.237 | 0.244 | 0.234 | 0.238 |

Note: Standard errors are clustered at the level of historical subnational regions (94 regions). Entries are unstandardized regression coefficients with standard errors in parentheses.

*$p < .05$

**$p < .01$

***$p < .001$

geographic and historical control variables suggested by the existing literature together with the country fixed effects. Even if some problems referring to causality still remain, our results might be interesting in light of correlational study as they illustrate the co-occurrence of authoritarian family relations and low out-group trust, which can mutually reinforce each other. In contrast, horizontally extended families were not shown to be an inherent feature of low-trust societies.

## 6. Conclusion

Our study examines the impact of historical family structures on today's level of out-group trust. Using national historical censuses for a set of European countries, we construct a broad dataset of indicators reflecting various dimensions of family organization in the past at the

**Table 5. OLS regression of contemporary out-group trust on generational hierarchy index, controlling for additional geographical and historical factors.**

| DV: out-group trust (LiTS, 2010) | (1) | (2) | (3) | (4) | (5) | (6) |
|---|---|---|---|---|---|---|
| Generational hierarchy | -0.730*** | -0.701*** | -0.776*** | -0.737*** | -0.738*** | -0.732*** |
|  | (0.194) | (0.179) | (0.205) | (0.197) | (0.198) | (0.196) |
| Gender hierarchy | 0.106 | 0.225 | 0.158 | 0.0292 | 0.0657 | 0.0489 |
|  | (0.292) | (0.264) | (0.337) | (0.290) | (0.274) | (0.265) |
| Mean HH | -0.0522 | 0.0435 | -0.166 | -0.125 | -0.125 | -0.140 |
|  | (0.212) | (0.170) | (0.210) | (0.192) | (0.205) | (0.196) |
| Population density | -2.15e-05 | 1.28e-05 | -1.33e-05 | -1.61e-05 | -2.55e-05 | -2.17e-05 |
|  | (5.01e-05) | (4.16e-05) | (5.37e-05) | (4.52e-05) | (4.68e-05) | (4.69e-05) |
| CWI | -1.050 | -1.674 | -1.604 | -1.518 | -1.211 | -1.274 |
|  | (1.161) | (1.196) | (1.260) | (1.109) | (1.203) | (1.048) |
| Caloric suitability index | 0.000254 |  |  |  |  |  |
|  | (0.000229) |  |  |  |  |  |
| Terrain ruggedness |  | -0.367** |  |  |  |  |
|  |  | (0.154) |  |  |  |  |
| Less than 50 km to the main river |  |  | 0.194 |  |  |  |
|  |  |  | (0.358) |  |  |  |
| Less than 50 km to the coast |  |  |  | -0.0972 |  |  |
|  |  |  |  | (0.123) |  |  |
| Prevalence of land suitability for hunting and gathering over agriculture |  |  |  |  | -0.190 |  |
|  |  |  |  |  | (0.942) |  |
| Medieval Catholic Church exposure |  |  |  |  |  | -0.0292 |
|  |  |  |  |  |  | (0.0315) |
| Individual level controls | YES | | | | | |
| Historical country FE | YES | | | | | |
| Observations | 4,042 | 4,042 | 4,042 | 4,042 | 4,042 | 4,042 |
| R-squared | 0.235 | 0.238 | 0.235 | 0.235 | 0.235 | 0.235 |

Note: Standard errors are clustered at the level of historical subnational regions (94 regions). Entries are unstandardized regression coefficients with standard errors in parentheses.

*$p < .05$

**$p < .01$

***$p < .001$

subnational level. We focus on the role of (a) family extendedness and (b) intra-family hierarchy in matters of gender and seniority.

Our findings are tested on a core sample of 94 regions within 8 Western and Eastern European countries (7 historical states). We show that family extendedness is not harmful to out-group trust. The most plausible theoretical explanation for this finding is that the "isolative" component of the extended family, associated with self-sufficiency and a low number of out-group contacts, counterbalances its "cooperative" component, which relies on impartial social norms owing to horizontal family extensions and permanent non-kin household members. We find that generational hierarchy within historical families has been a true obstacle for out-group trust up to today. This suggests that people from extended families might have avoided contact with out-groups not because they could satisfy all their needs within their large family, but because, being raised in an authoritarian family, they did not build enough trust to interact with the out-groups.

**Table 6. OLS regression of the contemporary level of out-group trust on generational hierarchy index components.**

| DV: out-group trust (LiTS, 2010) | unstd. beta | stand. beta |
|---|---|---|
| Prevalence of vertical hh headed by son | 4.269* | 0.09 |
| | (2.246) | |
| % vertical hh, all | -4.476*** | -0.399 |
| | (1.355) | |
| Gender hierarchy | 0.0991 | 0.045 |
| | (0.260) | |
| Mean HH size | 0.107 | 0.019 |
| | (0.190) | |
| Population density | -5.51e-05 | -0.031 |
| | (4.91e-05) | |
| CWI | -0.520 | -0.020 |
| | (1.032) | |
| Individual controls | YES | |
| Historical country FE | YES | |
| Observations | 4,042 | |
| R-squared | 0.233 | |

Note: Standard errors are clustered at the level of historical subnational regions (94 regions). Entries are unstandardized regression coefficients with standard errors in parentheses.

$^*p < .05$

$^{**}p < .01$

$^{***}p < .001.$

Moreover, our analysis suggests that only generational hierarchy has a detrimental effect on trust. This is a new finding that calls for more attention in future research on inter-generational relationships and the role of seniority in society.

Thinking about the broader implications of our study, we would suggest that the firm enculturation of authoritarian norms in pre-modern families was the real hindrance for the formation of trust and its positive consequences for institutional development, socio-economic modernization, and human emancipation. One should not overlook that at least in today's Europe, power relations within the family are qualitatively different from the authoritarian family order in the past. Complete obedience of younger generations to older ones and of women to men remains a dying attribute of large intergenerational family units. Contemporary family forms in general have become more liberal, which might lead to the lasting growth of out-group trust.

## Supporting information

**S1 File. Data sources and additional sample description.**
(DOCX)

**S2 File. Principal components analysis results.**
(DOCX)

**S3 File. OLS regressions results.**
(DOCX)

**S1 Fig. Validation of LiTS regional measure of out-group trust.**
(DOCX)

**S2 Fig. Vertical and lateral extensions across 94 regions.**
(DOCX)

**S3 Fig. Generational hierarchy and lateral extensions across 94 regions.**
(DOCX)

**S4 Fig. Vertical and lateral extensions excluding Albania.**
(DOCX)

**S5 Fig. Generational hierarchy and lateral extensions excluding Albania.**
(DOCX)

## Acknowledgments

We are grateful to Ulf Brunnbauer, Gilles Duranton, Siegfried Gruber, Karl Kaser, Sebastian Kluesner, Vladimir Kozlow, Aleksandr Libman, Jonathan Schulz, Peter Öri, Levente Pakot, and Franz Rothenbacher for their help with historical demographic data and useful comments on earlier versions of our paper.

## Author Contributions

**Conceptualization:** Maria Kravtsova, Aleksey Oshchepkov, Christian Welzel.

**Data curation:** Maria Kravtsova.

**Formal analysis:** Maria Kravtsova.

**Methodology:** Maria Kravtsova.

**Supervision:** Christian Welzel.

**Visualization:** Maria Kravtsova.

**Writing – original draft:** Maria Kravtsova, Aleksey Oshchepkov.

**Writing – review & editing:** Christian Welzel.

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
