## [Decision Letter · Decision Letter 0]

13 Jan 2023

PONE-D-22-15238THE SHADOW OF THE FAMILY: HISTORICAL ROOTS OF SOCIAL TRUST IN EUROPEPLOS ONE

Dear Dr. Kravtsova,

Thank you for submitting your manuscript to PLOS ONE. After careful consideration, we feel that it has merit but does not fully meet PLOS ONE’s publication criteria as it currently stands. Therefore, we invite you to submit a revised version of the manuscript that addresses the points raised during the review process.

I have taken over as the academic editor of this manuscript in December and could acquire three reviews from experts in the field on top of one already existing review that suggested minor revisions. All three new reviewers find your manuscript interesting, but all of them also point towards significant weaknesses that would have to be fixed before the manuscript can be fully evaluated and considered for publication in PLOS ONE. Please address all these suggestions either by implementing a meaningful solution or by providing a convincing rebuttal why you disagree with a suggestion. While you work on improving your manuscript, I would especially suggest that you make sure that it complies with all standards of academic writing. This is currently clearly not the case. Make sure that references are complete, up to date, and consistently formatted. It is surprising that one of your references will be published in 2029 and that is just to point out the most obvious mistake. Use a standard citation style. Schulz and Schulz et al., for example, are never treated as "the same author". Check that you are citing and discussing the relevant literature in the field. All tables and figures must be fully comprehensible without consulting the text. That means that notes should explain what numbers or other information shown in them reflect, if that is not already clear. Underlying methods of analysis, the use of subsamples etc. all need to be indicated such that the table/figure is fully comprehensible. Descriptive statistics and other information should be provided for the sample that is actually analyzed (at least in addition to the same information for a more comprehensive sample). That might already be the case, but the manuscript simply lacks many of these details. Variable names need to be as informative as possible. Figures cannot be placed in tables. If you want to add one figure composed of multiple maps label it as such, use consistent formatting and use one note below that figure. PLOS ONE has clear requirements for figures and figure files. In Figure 4, I would recommend using established and consistent abbreviations of country names, for example according to ISO standard. Provide informative and consistent axis labels in figures and eliminate uninformative elements, such as legends without information value. Do not place long web-links in the text of the paper, they are better relegated to a footnote or reference. This is not necessarily a complete list, so please make sure that the revised manuscript is of high quality in terms of academic writing standards, such that it can be properly evaluated. Also regarding the rest of the manuscript, please make sure that your presentation of all relevant information is concise and complete, such that the merits of your arguments and analysis can be properly evaluated. For example, please formulate your hypotheses more precisely. Is it on purpose that hypothesis 1 uses non-causal and hypothesis 2 uses causal language? Why is only the timing of the outcome but not that of the cause specified in every hypothesis?

We look forward to receiving your revised manuscript.

Kind regards,

Jerg Gutmann

Academic Editor

PLOS ONE

Journal Requirements:

5. We note that Figures 1-3 in your submission contain [map/satellite] images which may be copyrighted. All PLOS content is published under the Creative Commons Attribution License (CC BY 4.0), which means that the manuscript, images, and Supporting Information files will be freely available online, and any third party is permitted to access, download, copy, distribute, and use these materials in any way, even commercially, with proper attribution. For these reasons, we cannot publish previously copyrighted maps or satellite images created using proprietary data, such as Google software (Google Maps, Street View, and Earth). For more information, see our copyright guidelines: http://journals.plos.org/plosone/s/licenses-and-copyright.

 a. You may seek permission from the original copyright holder of Figures 1-3 to publish the content specifically under the CC BY 4.0 license. 

“I request permission for the open-access journal PLOS ONE to publish XXX under the Creative Commons Attribution License (CCAL) CC BY 4.0 (http://creativecommons.org/licenses/by/4.0/). Please be aware that this license allows unrestricted use and distribution, even commercially, by third parties. Please reply and provide explicit written permission to publish XXX under a CC BY li

Reviewers' comments:

Reviewer's Responses to Questions

**Comments to the Author**

1. Is the manuscript technically sound, and do the data support the conclusions?

Reviewer #1: Yes

Reviewer #2: Partly

Reviewer #3: Partly

Reviewer #4: Partly

2. Has the statistical analysis been performed appropriately and rigorously? 

Reviewer #1: Yes

Reviewer #2: No

Reviewer #3: N/A

Reviewer #4: Yes

3. Have the authors made all data underlying the findings in their manuscript fully available?

Reviewer #1: Yes

Reviewer #2: Yes

Reviewer #3: No

Reviewer #4: No

4. Is the manuscript presented in an intelligible fashion and written in standard English?

Reviewer #1: Yes

Reviewer #2: Yes

Reviewer #3: Yes

Reviewer #4: No

5. Review Comments to the Author

Reviewer #1: This paper is an interesting and well executed study on historical roots of social trust in Europe. I have two minor suggestions.

1. Mention the limitations of study, if any.

2. Compare the findings of the study with existing literature.

Reviewer #2: This is a well-written paper that provides an interesting analysis of how historical family ties may be related to current levels of out-group trust. The setup of the paper is clear, and the structure, in overall, sound, although some sections would require reordering as it becomes at times unclear what the key argument is.

The key finding of the paper is convincing, namely that household size does not appear to matter as much as hierarchical organization within the family. The paper uses interesting datasets and addresses important questions. However, I see one key problem in their research design:

The life in transition survey is representative at the national level, but the authors use single observations also at lower levels of aggregation. I do not think this is a correct procedure, in terms of statistics. If the authors used any additional computation to overcome such sample design problems or if they are in possession of further details that allow treating the data as statistically representative at the subnational level, they must provide details on it. Otherwise, the reader remains under the impression that it is not viable to use the LITs data at the subnational level at all.

The multilevel analysis in its current form has some important weaknesses.

First, the authors need to report the ICC of their null model to provide the reader with the relevant information of how much variability in the dependent variable is actually explained by the higher level included, so level 2 and level3. Similarly, results should report the R2 at the different levels, to make the reader understand which percentage of the contextual effect is explained by the level-specific control variables you include. Second, the authors use history as a level in which the present is nested. This is not uninteresting, but deserves some discussion - in particular because the individuals in the present (and their characteristics) are meant to be nested within the characteristics derived from other individuals in the past. Such design needs to be properly understood which does not seem to be the case when the authors discuss causality at the end of the results section. They use the typical disclaimer that results do not suggest causality, but in a multilevel model higher levels dominate lower levels, so the causal directionality is implicit. Further, that paragraph seems to forget that past family ties are put into connection with present levels of trust, which also only allows for one directionality of causality.

The inclusion of the third level - with only 7 observations - is rather problematic and appears to be a shortcut for the inclusion of control-variables only. It would be better to eliminate level 3 and to include relevant controls at level 2 - disaggregating controls at the right level where possible.

Minor comments:

• In the introduction it remains unclear for some time in the text whether "the network" refers to in and/or out although that specification comes a bit later

• Similarly, the text refers to extendedness at the start and may want to clarify from the start whether this is just household size or the typical concept of extended family

• the anticipated conclusion on page 5 lines 62-65 is a bit unclear - the correlation between the two covariates has not yet been adressed at that point of the manuscript

• unclear for which reason the extended sample is introduced in the introduction if it does not serve the purpose of the manuscript (76-78) - could be made clear it is only used for robustness checks

• page 8 - 125-127 unclear

• neolocality in table 1 seems wrong - shouldn't it be more neolocality when the share of youngsters living only with their spouse is higher?

• On page 15 the description of the gender hierarchy index appears to mix indicators with opposite directionality (e.g. married women at young age and married women at older age) this is misleading and is actually in contrast with the specifics given in table 1 - authors could add at least a footnote assuring the reader that all directionalities are expected and to see table 1 for details - so phrase currently in lines 313-314 may be anticipated somehow

• Figure 4 and the entire treatment of "pure" horizontal and vertical linkage in which however horizontal ones appear to be comprised is unconvincing and seems to lead the reader astray. If vertical linkage comprise horizontal then this graph is not very helpful and the analysis dedicated to it somehow misleading.

• page 21: 432 - why is a weak correlation a sign for complementarity instead of substitutability? This should be derived from the sign, not from the strength of the correlation.

• the deepest explanation of isolative vs. cooperative comes only in the conclusion. Better treatment of the point should be anticipated in the manuscript.

Reviewer #3: Referee Report for PONE-D-22-15238

THE SHADOW OF THE FAMILY: HISTORICAL ROOTS OF SOCIAL TRUST IN EUROPE

The paper tests whether elements of family arrangements affect out-group trust. Using a sample of seven historical countries they establish that generational hierarchy within the household has a significant effect on trust whereas family extendedness does not.

The results are interesting and there is an extensive analysis of the intuition behind the analysis and the surprising result. I feel however that the empirical implementation is not as strong as it should be to convince about the finding. Below I provide a list of suggestions related to the empirical part and the overall framing of the paper.

Framing of the paper

1. The first paragraphs do not frame the paper very well. Is it a paper on globalization, comparative development, as indicted in the first two paragraphs, or a paper for trust as indicated in the appendix?

2. As the overall paper suggests it is about trust, the paper could as well start with the third paragraph. I think everyone would agree that trust is indeed a crucial element for development. No need to frame that already in the intro.

3. Crucially I do not see any reference to the papers of Enke who has extensively worked on the empirics of those topics (see e.g., Kinship, Cooperation, and the Evolution of Moral Systems). The papers and the associated controls should be accounted for, in the literature review and empirically as well.

4. Overall, as also evident in the intro and the lit. review the paper should clearly make the distinction between family ties and family types. To discuss what exactly is the central point in the paper and to make it clear throughout. What is exactly that the authors have in mind when they say family arrangements. They discuss the two axes of the arrangement, i.e., family extendedness and hierarchical relations”, but in which exact category does this fall in? This will help them refine their argument and benchmark their contribution better.

5. Does the no of people in the household capture family extendedness or just fertility? How can we know who lives in this extended households, i.e., its composition?

6. Some editing would be useful before resubmitting.

Comments on the Empirics

1. A major concern is the sample selected. As it includes a set of transition countries, whose family structures were challenged by the institutions, I am wondering about the external validity of the argument. In different places, where the family setting was not as challenged, what are the results. Can you benchmark your results in a different sample or from a different paper?

2. I do not really see why the authors are not able to control for country fixed effects, historical or contemporary. The fact that they only have 7 historical countries does not prevent them to do so to the extend they have within country variation. I would thus feel more comfortable if they included a table with historical and current country fixed effects even in a robustness section, given that the set of controls they include is extremely limited.

3. I would also like to see the results with clustering standard errors at the subnational level.

4. In terms of formatting I would like to see much better tables, much more informative and well structured. E.g., Enke whom I mentioned already above is an example.

5. The empirical part should be much better drafted in terms of content as well, e.g., discuss further what the actual magnitude of the coefficients is.

6. As additional light on the results, It would also be interesting presenting separately the results for the subindices of generational hierarchy to figure out which particular element drives the results.

7. In the presence of unobservables it would be useful undertaking the Oster test for omitted variables.

Reviewer #4: The paper is interested in determining the potential effects of various aspects of historical family organization on social trust. It adds to a wave of relatively recent papers that are interested in determining a number of long-run effects of historical family organization. Although it is part of this wave, I think the paper suffers from a number of weaknesses, which is why I recommend not to publish it in PlosOne before having undergone a major revision.

Let me briefly explain:

I think the theoretical arguments described here need to better developed and could be moved closer to the arguments already described in the recent literature just mentioned. I am, e.g., not sure if it is true that previous literature was merely concerned with the nuclearity/extendedness dimension (as claimed on line 56f.). I always read the number of generations living under one roof – as analyzed by Todd – as including an aspect of hierarchy. This (imprecise – in my point of view) position is then repeated in lines 174ff. The addition (lines 196f.) that horizontal family additions should be beneficial to out-group trust is included without a precise argument.

At times, one gets the impression that the authors have only superficial knowledge of the literature. The book cited by Todd, e.g. is not his most relevant contribution to the argument as such. In my mind, it would make more sense to cite his first book on the issue (The Explanation of Ideology).

Among the more recent papers, there are some that deal with questions closely related to the one dealt with in this paper. E.g., Gutmann & Voigt (2022) is a paper testing many of Todd’s predictions, one of them being xenophobic or racist attitudes – which should be closely related with out-group-trust.

In the context of hypo 2b, it would have been nice to take Alesina et al (2013; plough) into account and explicitly control for the suitability of land.

Although the data-collection effort is laudable, the description of the variables needs to be improved (e.g. lines 275f. are incomprehensible as is). The argument behind the coding of the vertical extension did not convince me. With regard to group 5, it needs to be made clear in how far the various aspects are indicative of gender (in-)equality.

The econometric models in 299ff. are also in need of a better explanation

Minor comments:

In lines 86ff., three studies are quoted that deal with historical determinants of social trust. These are well-known studies but there are many more. Why these? The choice seems to be pretty arbitrary.

I am not sure I understand the order in which references are presented. It seems that it is neither the alphabetical order of the authors, nor the years in which the papers were published (see, e.g., lines 27f.)

6. PLOS authors have the option to publish the peer review history of their article (what does this mean?). If published, this will include your full peer review and any attached files.

Reviewer #1: No

Reviewer #2: No

Reviewer #3: No

Reviewer #4: No

---

## [Author Response · Author response to Decision Letter 0]

4 Apr 2023

Comments from the editor:

While you work on improving your manuscript, I would especially suggest that you make sure that it complies with all standards of academic writing. This is currently clearly not the case. Make sure that references are complete, up to date, and consistently formatted. It is surprising that one of your references will be published in 2029 and that is just to point out the most obvious mistake. Use a standard citation style. Schulz and Schulz et al., for example, are never treated as "the same author". Check that you are citing and discussing the relevant literature in the field. 

We double-checked all the references mentioned in the paper. 

All tables and figures must be fully comprehensible without consulting the text. That means that notes should explain what numbers or other information shown in them reflect, if that is not

already clear. Underlying methods of analysis, the use of subsamples etc. all need to be indicated such that the table/figure is fully comprehensible. 

We did our best to produce understandable tables and figures

Descriptive statistics and other information should be provided for the sample that is actually analyzed (at least in addition to the same information for a more comprehensive sample). That might already be the case, but the manuscript simply lacks many of these details. 

We added descriptive statistics for our core sample (94 regions) in addition to the same information for a more comprehensive sample (170 regions). Descriptive statistics for the core sample was placed in Appendix, while the reference to this information is given in the text (page 22). Table 3 contains correlations between family extendedness indicators and hierarchy measures for the core sample (94 regions) and for a larger sample (170 regions). 

Variable names need to be as informative as possible.

We tried to make the variable names as informative as possible. 

Figures cannot be placed in tables. If you want to add one figure composed of multiple maps label it as such, use consistent formatting and use one note below that figure. PLOS ONE has clear requirements for figures and figure files. In Figure 4, I would recommend using established and consistent abbreviations of country names, for example according to ISO standard.

We removed our figures from the tables and placed them in separate files. In Figure 4 we don’t use abbreviations of country names any more. We replaced them with different symbols and the full country names are given in the legend.

 Provide informative and consistent axis labels in figures and eliminate uninformative elements, such as legends without information value. 

We provided axis names and removed uninformative elements placing additional information as a note under the figures.

Do not place long web-links in the text of the paper, they are better relegated to a footnote or reference. This is not necessarily a complete list, so please make sure that the revised manuscript is of high quality in terms of academic writing standards, such that it can be properly evaluated.

Long web-links were placed in a footnote.

Also regarding the rest of the manuscript, please make sure that your presentation of all relevant information is concise and complete, such that the merits of your arguments and analysis can be properly evaluated. For example, please formulate your hypotheses more precisely. Is it on purpose that hypothesis 1 uses non-causal and hypothesis 2 uses causal language? Why is only the timing of the outcome but not that of the cause specified in every hypothesis?

We reformulated our hypotheses in a more uniform and appropriate way.

Reviewer #1: 

This paper is an interesting and well executed study on historical roots of social trust in Europe. I have two minor suggestions.

1. Mention the limitations of study, if any.

The insightful comments of all reviewers helped us to better understand existing limitations of our study. We discuss them explicitly in the revised version at the end of the results section.

2. Compare the findings of the study with existing literature.

We did our best to place our findings into the existing literature. Our results are best comparable with studies that examine the effect of the nuclear family on societal outcomes. Using our dataset, we confirm that the nuclear family is beneficial for the formation of out-group trust (Duranton et al., 2009; Enke, 2019; Gutmann & Voigt, 2022; Schulz, 2019b). Additionally, we provide new evidence that this effect exists not just due to a smaller family size but rather due to the lack of generational hierarchy. In extended families, in contrast, the dominance of older generations over younger prevents trust formation. 

Reviewer #2: 

This is a well-written paper that provides an interesting analysis of how historical family ties may be related to current levels of out-group trust. The setup of the paper is clear, and the structure, in overall, sound, although some sections would require reordering as it becomes at times unclear what the key argument is. The key finding of the paper is convincing, namely that household size does not appear to matter as much as hierarchical organization within the family. The paper uses interesting datasets and addresses important questions. However, I see one key problem in their research design:

The life in transition survey is representative at the national level, but the authors use single observations also at lower levels of aggregation. I do not think this is a correct procedure, in terms of statistics. If the authors used any additional computation to overcome such sample design problems or if they are in possession of further details that allow treating the data as statistically representative at the subnational level, they must provide details on it. Otherwise, the reader remains under the impression that it is not viable to use the LITs data at the subnational level at all.

We agree that this is a methodological limitation of our study, which we now mention in the revised version (see footnote 10 at p.19). However, we believe that the non-representativeness of LiTS at the subnational level it is not a crucial issue in our case as we don’t use LiTS data to construct any indicators at the subnational level. Rather, we assign historical indicators taken from external data sources to all individuals living within the same primary sampling units (PSU). In this case, a two-stage quasi-random sampling procedure used in LiTS (first, for each country 50-75 PSUs are chosen from a larger country-specific list of PSUs with probability of selection proportional to the PSU size, and, second, households are randomly chosen within each PSU) should prevent any systematic bias in the composition of PSU samples and thus should guarantee that our estimated coefficients are free of the sample-selection bias. The representativeness of LiTS at subnational level would be a pleasant bonus in this case, but it shouldn’t affect the econometric estimates.

Additionally, we would like to note that the literature contains plenty published examples which follow a similar procedure as we do: first, match subnational level indicators to microdata from cross-national surveys which are not representative at the subnational level, and, second, estimate the impact of those indicators on some individual level dependent variables. For instance, Ajzenman et al., (2022) in their recent study also use LiTS data to show that the exposure to mass transit migration measured at the PSU level (the distance from the locality to the closest transit migrant route) affects individual entrepreneurial activity and anti-migration sentiments. Schulz et al., (2019) assigns historical indicators calculated at the level of contemporary subnational regions to the European Social Survey (ESS) data which are not representative at the subnational level as well. Duranton et al. (2009) derive their measures of social capital and income inequality at the NUTS-2 level from the European Community Household Panel (ECHP) microdata, which are not representative at the subnational level. This last example, probably, is vulnerable to the problem of the non-representativeness at the subnational level, but this is not the case of our paper.

The multilevel analysis in its current form has some important weaknesses.

We emphasize that in the revised version we changed our estimation approach from multilevel modeling (MLM) to OLS regressions with country fixed effects (FE) and cluster-robust estimation of standard errors of coefficients (CRSE). This allows us to avoid some methodological concerns (e.g., the low number of observations at the third level) and to address the comments of Reviewer #3. This change from MLM to OLS-CRSE with FE is also justified by the fact that MLM is much less applied in the economic literature than OLS-based methods (see Oshchepkov & Shirokanova, 2022). The main estimated model is now presented by Equation (1) in the revised text.

First, the authors need to report the ICC of their null model to provide the reader with the relevant information of how much variability in the dependent variable is actually explained by the higher level included, so level 2 and level3. 

As we moved to OLS-CRSE instead of MLM, the null model and the corresponding ICC parameter lose their value and meaning within the OLS-based framework. 

Similarly, results should report the R2 at the different levels, to make the reader understand which percentage of the contextual effect is explained by the level-specific control variables you include. 

As we moved to OLS-CRSE instead of MLM, we follow the standard practice to report the overall R2 after every regression in each table with econometric results.

Second, the authors use history as a level in which the present is nested. This is not uninteresting, but deserves some discussion - in particular because the individuals in the present (and their characteristics) are meant to be nested within the characteristics derived from other individuals in the past. Such design needs to be properly understood which does not seem to be the case when the authors discuss causality at the end of the results section. They use the typical disclaimer that results do not suggest causality, but in a multilevel model higher levels dominate lower levels, so the causal directionality is implicit. Further, that paragraph seems to forget that past family ties are put into connection with present levels of trust, which also only allows for one directionality of causality.

Despite the fact that now we apply OLS-CRSE instead of MLM, this comment remains very important for us. We are completely agreeing that there are good prerequisites to think that the statistically significant correlation between out-group trust and regional historical family indicators established in our empirical analysis reflects the causal impact of historical family on trust. Firstly, the subnational level clearly dominates the individual level, i.e., out-group trust of single individuals cannot affect the family structure in the region. Secondly, family structures are measured in the past and thus they cannot be affected by present levels of trust. However, there is still may be a room for endogeneity concerns due to the omitted variable which affects both out-group trust and historical family arrangements. Therefore, we are still cautious in our claims of causality, though we clarified and improved the corresponding discussion in the revised version of the text.

The inclusion of the third level - with only 7 observations - is rather problematic and appears to be a shortcut for the inclusion of control-variables only. It would be better to eliminate level 3 and to include relevant controls at level 2 - disaggregating controls at the right level where possible.

Following OLS-CRSE estimation methodology we include historical country FE, which helps to avoid the critics related to the low number of observations at the third level, inherent to MLM.

Minor comments:

In the introduction it remains unclear for some time in the text whether "the network" refers to in and/or out although that specification comes a bit later

Following this comment and also comments of other reviewers, we revised the introduction. 

Similarly, the text refers to extendedness at the start and may want to clarify from the start whether this is just household size or the typical concept of extended family

We added a special footnote (footnote 6 at p.4) to clarify that the term “extendedness” reflects in this case just the number of relatives in the household. It should not be confused with P. Laslet’s (1983) term “extended family unit” which stands for a household with single relatives who are not included in family nucleus. 

the anticipated conclusion on page 5 lines 62-65 is a bit unclear - the correlation between the two covariates has not yet been addressed at that point of the manuscript

We decided not to change these sentences because they make our results clear from the introduction. This allows the reader to know about our findings without reading the whole paper. 

unclear for which reason the extended sample is introduced in the introduction if it does not serve the purpose of the manuscript (76-78) - could be made clear it is only used for robustness checks

We revised the introduction and removed mentioning the extended sample from the introduction and made it clear in the data description section that we use this sample only for descriptive analysis and robustness checks. 

 page 8 - 125-127 unclear

This paragraph was improved in the revised version.

neolocality in table 1 seems wrong - shouldn't it be more neolocality when the share of youngsters living only with their spouse is higher?

You are completely right: higher share of youngsters living only with their spouse means higher neolocality. We borrowed this indicator from Gruber and Szoltysek (2016) and it was a composite part of their Patriarchy index. The authors rescaled all the items to a range from 0-10 so that higher values mean higher patriarchy. We had to use Gruber and Szoltysek’s rescaled items instead of the original proportions of youngsters living with their spouse and children because they were not available. Therefore, higher scores on neolocality mean more patriarchy, and correspondingly a lower proportion of youngsters living with his spouse. We are very sorry that our description of this variable was misleading.

However, we decided to exclude this item from the current version of the generational hierarchy index. We realized that this item measures not only vertical extensions but partially also the horizontal extensions. If a young man lives not only with his spouse but also with other relatives it does not mean that his household is vertically extended. It might be the case that he lives with his lateral relatives like siblings, aunts, cousins etc. For the same reason we excluded from our generational hierarchy index also MUH (Marital units per household) measure that considers all marital units (MU) including MU formed by horizontal extensions. The construction method of our new generational hierarchy index is thoroughly described in the text. We believe that it became more transparent.

On page 15 the description of the gender hierarchy index appears to mix indicators with opposite directionality (e.g. married women at young age and married women at older age) this is misleading and is actually in contrast with the specifics given in table 1 - authors could add at least a footnote assuring the reader that all directionalities are expected and to see table 1 for details - so phrase currently in lines 313-314 may be anticipated somehow

Thank you very much for mentioning this point. Actually, in the previous version all gender hierarchy items were rescaled so that higher scores mean more gender inequality. We are sorry that it was not so clear to the reader. In the current version of our paper we completely revised our indices. We now draw the first principal component from all the index components instead of summing up the items scores. We believe that principal components analysis is a more adequate method of index composition in our case where we have several similar items. 

In the current version of the gender hierarchy index four items coming from Gruber and Szoltysek’s (2016) patriarchy index (% of female household heads, % of married women 15-19, % of wives who are older than their husbands, % of young women 20-34 who live in non kin families) were originally rescaled so that higher scores mean more gender inequality. These items affect the factor score of gender hierarchy positively. The percentage of never married women aged 20-29 which is the proxy for higher age at first marriage affects the factor score negatively. We make it clear in our text.

Figure 4 and the entire treatment of "pure" horizontal and vertical linkage in which however horizontal ones appear to be comprised is unconvincing and seems to lead the reader astray. If vertical linkage comprise horizontal then this graph is not very helpful and the analysis dedicated to it somehow misleading.

We are grateful for this remark. We believe it makes sense to plot the proportion of all vertically extended households and the proportion of all horizontally extended households. This is what we do in the current version of the paper. If we plot vertical extensions excluding horizontal and horizontal extensions excluding vertical we loose a lot of information. For example, in Albania most households have both vertical and horizontal extensions. Therefore, it would have low scores on both mutually exclusive indicators of family extensions though most households in Albania are extended.

page 21: 432 - why is a weak correlation a sign for complementarity instead of substitutability? This should be derived from the sign, not from the strength of the correlation.

Following the comments from our reviewers we revised our indices to make them more transparent. In course of this procedure the correlation has been changed so that this sentence is not actual any more. 

the deepest explanation of isolative vs. cooperative comes only in the conclusion. Better treatment of the point should be anticipated in the manuscript.

Thank you for the suggestion, we agree that it will clarify our argument. We introduce the explanation of the “isolative” and “cooperative” components in the introduction and then we elaborate it in the hypotheses section.

Reviewer #3: 

The paper tests whether elements of family arrangements affect out-group trust. Using a sample of seven historical countries they establish that generational hierarchy within the household has a significant effect on trust whereas family extendedness does not.

The results are interesting and there is an extensive analysis of the intuition behind the analysis and the surprising result. I feel however that the empirical implementation is not as strong as it should be to convince about the finding. Below I provide a list of suggestions related to the empirical part and the overall framing of the paper.

Framing of the paper

1. The first paragraphs do not frame the paper very well. Is it a paper on globalization, comparative development, as indicted in the first two paragraphs, or a paper for trust as indicated in the appendix?

2. As the overall paper suggests it is about trust, the paper could as well start with the third paragraph. I think everyone would agree that trust is indeed a crucial element for development. No need to frame that already in the intro.

Following this comment, we edited and polished the introduction and removed the first two paragraphs.

3. Crucially I do not see any reference to the papers of Enke who has extensively worked on the empirics of those topics (see e.g., Kinship, Cooperation, and the Evolution of Moral Systems). The papers and the associated controls should be accounted for, in the literature review and empirically as well.

We acknowledge this omission and cite the important contribution of B. Enke in the revised version. Moreover, we introduce a variable that measures ‘land suitability for hunting and gathering’ (Beck & Sieber, 2010) into the set of our controls at the subnational level. B. Enke mentions that this indicator may have a crucial impact on kinship systems and uses a similar measure based on Murdock’s Ethnographic Atlas in his models. 

4. Overall, as also evident in the intro and the lit. review the paper should clearly make the distinction between family ties and family types. To discuss what exactly is the central point in the paper and to make it clear throughout. What is exactly that the authors have in mind when they say family arrangements. They discuss the two axes of the arrangement, i.e., family extendedness and hierarchical relations”, but in which exact category does this fall in? This will help them refine their argument and benchmark their contribution better.

It is, indeed, a very interesting question. Family extendedness and hierarchical relations fall definitely into the category of family types. In our point of view strong family ties imply great importance of the family, readiness to sacrifice their own interests for the family, and perceived obligation to render assistance to their relatives. Alesina, Giuliano, 2014 (Family ties) suggest that extended families are correlated with strong family ties, however Banfield’s ‘amoral familism’ refers to nuclear families in the Southern Italy. Intuitively it is not obvious that people living in extended families attach greater importance to family and are ready to sacrifice for it. On the one hand extended families are associated with more collectivism and obedience but on the other hand a large number of relatives with different interests provokes a lot of conflicts within an extended household that may weaken family ties. We believe that the relationship between family ties and family types deserves a special study. We highlight in the text that our paper is devoted to the effect of family types in terms of the number of people and hierarchical relations. Our hypotheses rely on these characteristics while family ties would require additional mechanisms and additional hypotheses. 

5. Does the no of people in the household capture family extendedness or just fertility? How can we know who lives in this extended households, i.e., its composition?

Thank you for this remark. We are sorry that we did not make it clear enough in the previous version of the text. We rewrote this section to make it more accessible to the reader. Our core sample is based on census individual level data coming from IPUMS International (https://international.ipums.org/international/) and the Mosaic project (https://censusmosaic.demog.berkeley.edu/). These data collections contain information for each individual on age, gender, marital status and relation to the household head. It allowed us to construct different family structure indicators. The number of people in the household comes from two sources. For some countries we created this indicator using the individual level data while for some countries we used aggregates from the printed sources. The no of people in the household captures both family extendedness and fertility. Therefore, we control for the children/adults ratio in the model specifications where we use this general indicator. Additionally, we include in our models the more precise measures of household extendedness like “the proportion of households comprising lateral relatives (aunts, uncles, cousins etc.) that do not have vertical extensions (parents of the household head, adult children, grandchildren etc.), or the mean household size without children aged 0-14. 

6. Some editing would be useful before resubmitting.

We did our best in editing the text.

Comments on the Empirics

1. A major concern is the sample selected. As it includes a set of transition countries, whose family structures were challenged by the institutions, I am wondering about the external validity of the argument. In different places, where the family setting was not as challenged, what are the results. Can you benchmark your results in a different sample or from a different paper?

It is a really important concern. The sample selection is associated mostly with the availability of historical data which may cause questions about the generalizability of our results. We agree that more studies are needed to test whether our results can be replicated on a larger sample of countries. 

Indeed, our sample is rather skewed towards the West European countries (France, Great Britain and Sweden) as we have for them 73 (out of 94) regional level observations. The reason is that the West European countries have better historical statistics than the East European countries. Therefore, most of the regions in our sample did not experience major changes in family setting due to the Communist past. 

Our results are generally in line with the results of Duranton et al., (2009) who conducted regional level analysis for the West European countries. 

2. I do not really see why the authors are not able to control for country fixed effects, historical or contemporary. The fact that they only have 7 historical countries does not prevent them to do so to the extend they have within country variation. I would thus feel more comfortable if they included a table with historical and current country fixed effects even in a robustness section, given that the set of controls they include is extremely limited.

Let us briefly explain that the multilevel modeling (MLM) framework assumes that all regional and country level effects are random (RE). To take into account possible omitted variables at the country level one uses centering subnational variables around the country means, as it was mentioned in lines 368-376 of the previous version of the text. This technique is equivalent to including country FE (e.g., see Bell et al., 2019; Oshchepkov & Shirokanova, 2022).

In the revised version, we made our estimation approach more straightforward and changed from MLM to OLS regressions with country fixed effects (FE) and cluster-robust estimation of standard errors of coefficients (CRSE). The main estimated model is now presented by Equation (1) (lines 362-363 in the revised version). This methodological change directly addresses the issue of (non)inclusion of country FE.

Please, note that we include historical country FE but not FE for the contemporary countries to better control for omitted variables which may affect both historical family structures and out-group trust. Obviously, contemporary societal characteristics don’t affect historical family structures but rather may be influenced by them.

3. I would also like to see the results with clustering standard errors at the subnational level.

In the previous version of the paper, a possible correlation of errors within subnational units was taken into account within the MLM framework by introducing random effects at the subnational level (see Oshchepkov & Shirokanova, 2022). In the revised version, we apply cluster robust estimation of standard errors (CRSE), as suggested. 94 observations at the subnational level guarantee that these estimates are consistent. 

4. In terms of formatting I would like to see much better tables, much more informative and well structured. E.g., Enke whom I mentioned already above is an example.

We did our best to make our tables more comprehensive. 

5. The empirical part should be much better drafted in terms of content as well, e.g., discuss further what the actual magnitude of the coefficients is. 

We completely agree that the magnitude of the coefficients is important and added the corresponding discussion to our results section. See lines 524-525, 534-535.

6. As additional light on the results, It would also be interesting presenting separately the results for the subindices of generational hierarchy to figure out which particular element drives the results.

Thank you very much for this suggestion. After doing this exercise we decided to revise completely our generational hierarchy index. We realized that the results were driven by the proportion of vertically extended households (comprising adult children 20+) and by the share of households comprising three generations headed by the son instead of his father. This indicator means less generational hierarchy compared to the vertically extended households headed by the oldest man in the household. It is revealed to be positively linked with out-group trust.

We decided to remove from our index MUH (marital units per household) and neolocality that do not have a significant effect on out-group trust and measure not only vertical but also horizontal family extensions. Neolocality measures the proportion of youngsters who live only with his spouse and small children. However, if a young man lives not only with his spouse but also with other relatives it does not mean that his household is vertically extended. It might be the case that he lives with his lateral relatives like siblings, aunts, cousins etc. MUH has the same problem as it considers all marital units (MU) including MU formed by horizontal extensions. We also removed from the generational hierarchy index one of the two items measuring in different ways whether the multigenerational households were headed by representatives of the older or of the younger generations.

The current version of generational hierarchy index captures whether 1) a household is vertically extended, 2) whether a vertically extended household is headed by the son or by his father. Firstly, we assume that nuclear families that live separately from their parents are less influenced by the older generations compared to the vertically extended families. Secondly, vertically extended families headed by the father have higher generational hierarchy compared to the vertically extended households headed by the son. 

The construction of both index components is thoroughly described in the text. As a robustness check we include them separately in the core specification of our regression (see table 6). Both of them are significant. The prevalence of vertically extended households headed by son has a positive effect on out-group trust while the proportion of vertically extended households affects out-group trust negatively.

7. In the presence of unobservables it would be useful undertaking the Oster test for omitted variables.

We agree that the Oster’s test (Oster, 2019) may serve as a useful tool to diagnose the omitted variable problem and assess the quality of controls. We performed this test for our core specification of Equation (1) that includes the following set of controls: individual level controls, Cool Water Index capturing temperature and precipitations, historical population density and historical country FE. Assuming that Rmax = 1.3*Rfull , as suggested by Oster (2019), we have got the test’s statistics delta less than 1, which suggests that selection on unobservable is greater than selection on observables in our case, thus it would be good to include more or better controls. With this aim, we tried a few additional control variables at the level of historical regions, as suggested by previous historical studies: the caloric suitability of land for agriculture, terrain ruggedness, proximity to the waterways, land suitability for hunting and gathering and exposure to the Medieval Catholic church. The inclusion of these variables neither changed our key findings nor significantly improved the value of delta. As we are not aware of any other theoretically useful historical/geo-climatic controls that could be used in our regressions, we believe that our estimates results are not prone to the omitted variable problem. 

Reviewer #4: 

The paper is interested in determining the potential effects of various aspects of historical family organization on social trust. It adds to a wave of relatively recent papers that are interested in determining a number of long-run effects of historical family organization. Although it is part of this wave, I think the paper suffers from a number of weaknesses, which is why I recommend not to publish it in PlosOne before having undergone a major revision. 

Let me briefly explain:

I think the theoretical arguments described here need to better developed and could be moved closer to the arguments already described in the recent literature just mentioned. I am, e.g., not sure if it is true that previous literature was merely concerned with the nuclearity/extendedness dimension (as claimed on line 56f.). I always read the number of generations living under one roof – as analyzed by Todd – as including an aspect of hierarchy. 

We are really grateful for this comment. In fact, nucleary/extendedness dimension overlap in some points with generational hierarchy. For this reason, it is mistakenly to say that previous literature ignored generational hierarchy. We are sorry, that we did not make it clear in the previous version of the manuscript. Now this point is clarified in the text. Nuclear family is by definition less hierarchical because the younger generations live separately from the older generations. It lessens considerably parents’ authority over their adult offspring. In case of the extended family the situation is less unambiguous. Extended families may imply adult children remaining in their parental household upon marriage, as well as a number of lateral relatives (siblings, aunts, uncles, nephews, nieces, cousins) with their families forming an extended household. In the former case we could speak about generational hierarchy while in the latter case we deal with a simple two generational household as lateral relatives are of the same generation as the members of the family nucleus. Moreover, co-residence of parents and their adult children does not necessarily mean the dominance of older generations over younger. Often such households are headed by the representatives of the younger generation while parents have a dependent status. Obviously, these households cannot be considered as patriarchal families where older generations have a well-established power over younger.

Existing data collections on family structure did not allow to disentangle family extendedness in terms of the number of relatives from generational hierarchy. Fortunately, we can benefit from historical census microdata for individuals and create more precise indicators. Therefore, we can answer an additional question that has not been answered in the existing literature: why are nuclear family arrangement beneficial for out-group trust? Is it what matters the effect of small family size and the objective need to cooperate with people beyond the kin group or is it the lack of strong hierarchical relations preventing trust formation? 

This (imprecise – in my point of view) position is then repeated in lines 174ff. The addition (lines 196f.) that horizontal family additions should be beneficial to out-group trust is included without a precise argument.

Thanks for this remark. We tried to make this point more precise in the text. Putnam’s (1993) idea was that when people of equal social status interact on a daily basis with each other they elaborate stronger norms of reciprocity and perceive each other as more trustworthy. Horizontal family extensions imply a larger number of relatives of the same generation and, accordingly, of equal power status within the family who interact with each other every day. According to Putnam it may stimulate trust formation. When people learn to trust each other within the family during their formative years they may be more likely to trust also unknown people or people of other nationality and religion. 

At times, one gets the impression that the authors have only superficial knowledge of the literature. The book cited by Todd, e.g. is not his most relevant contribution to the argument as such. In my mind, it would make more sense to cite his first book on the issue (The Explanation of Ideology).

Thank you for the comment. “The Explanation of Ideology” is in fact the most cited Todd’s paper. However, we cited not arbitrarily “L’invention de l’Europe”. We use in our paper Todd’s map digitalized by Duranton et al., (2009) for comparison. This map was borrowed from “L’invention de l’Europe” and it has only 5 family types, while in “The Explanation of Ideology” there are 7 family types (Todd distinguishes additionally between endogamy and exogamy). Given that “The Explanation of Ideology” might be more familiar to the reader we added a citation of this book. 

Among the more recent papers, there are some that deal with questions closely related to the one dealt with in this paper. E.g., Gutmann & Voigt (2022) is a paper testing many of Todd’s predictions, one of them being xenophobic or racist attitudes – which should be closely related with out-group-trust.

Thanks for this reference, it is a really very interesting recent paper that is closely related to our analysis. We added it to our literature review. 

In the context of hypo 2b, it would have been nice to take Alesina et al (2013; plough) into account and explicitly control for the suitability of land.

Thank you for this suggestion, we include land suitability for agriculture as a control variable and cite Alesina et al., (2013) in the revised version.

Although the data-collection effort is laudable, the description of the variables needs to be improved (e.g. lines 275f. are incomprehensible as is). The argument behind the coding of the vertical extension did not convince me. 

It is a very important remark. We tried to make the description of our variables more comprehensive. Additionally, we completely revised our variables and the methodology of index construction. We created four variables that reflect different aspects of vertical family extensions. These are 1) the proportion of all vertically extended households (comprising parents, parents in law, adult or married children, children in law, (great) grandchildren; 2)proportion of vertically extended households headed by father; 3) proportion of vertically extended households headed by son (comprising a person whose relationship to the household head is “parent” and gender is “male”); 4)prevalence of vertically extended households headed by son over vertically extended households headed by father. 

To create the current version of the generational hierarchy index we draw the first principal component from two items 1) the proportion of all vertically extended households; 2) prevalence of vertically extended households headed by son. This index is based on two assumptions. Firstly, nuclear families that live separately from their parents are generally less influenced by the older generations compared to the vertically extended families. Secondly, vertically extended families headed by the father have higher generational hierarchy compared to the vertically extended households headed by the son. 

We also tried to make our generational hierarchy index more transparent removing several items. We removed MUH (marital units per household) and neolocality that measure not only vertical but also horizontal family extensions. Neolocality measures the proportion of youngsters who live only with his spouse and small children. However, if a young man lives not only with his spouse but also with other relatives it does not mean that his household is vertically extended. It might be the case that he lives with his lateral relatives like siblings, aunts, cousins etc. MUH has the same problem as it considers all marital units (MU) including MU formed by horizontal extensions. We also removed from the generational hierarchy index one of the two items measuring in different ways whether the multigenerational households were headed by representatives of the older or of the younger generations.

With regard to group 5, it needs to be made clear in how far the various aspects are indicative of gender (in-)equality.

We improved the description of variables that constitute the gender hierarchy index. We tried to explain how each of those variable is related to gender (in)equality.

The econometric models in 299ff. are also in need of a better explanation

Please, not that in the revised version we changed our estimation approach from multilevel modeling (MLM) to OLS regressions with country fixed effects (FE) and cluster-robust estimation of standard errors of coefficients (CRSE). This allows us to avoid some methodological concerns and to address the comments of Reviewer #2 and Reviewer #3. The main estimated model is now presented by Equation (1) (see lines 362-363 in the revised text). We hope that now our econometric methodology is more straightforward and clear.

Minor comments:

In lines 86ff., three studies are quoted that deal with historical determinants of social trust. These are well-known studies but there are many more. Why these? The choice seems to be pretty arbitrary.

Of course, trust and its determinants is an extremely popular topic in social sciences and there are many excellent papers that examine historical roots of trust. Unfortunately, the comprehensive review of that literature deserves a separate study and out of the scope of our paper, as we are focused on one specific determinant of trust, namely family organization. Therefore, we just mention a few previous papers on the historical roots of trust, which, in our point of view, are among the most influential and creative, and then move to the discussion of literature which is more related to our paper. In order to smooth that impression of arbitrariness we slightly corrected a way how we refer to those three studies.

I am not sure I understand the order in which references are presented. It seems that it is neither the alphabetical order of the authors, nor the years in which the papers were published (see, e.g., lines 27f.)

We fixed that inconsistency and now present the references in the alphabetical order.

References used in the cover letter:

Bell, A., M. Fairbrother, and K. Jones. Fixed and Random effects models: making an informed choice. Quality & Quantity. 2019; 53(2): 1051-1074.

Oshchepkov, A. & A., Shrokanova. Bridging the gap between multilevel modeling and economic methods. Social Science Research. 2022; 104: 102689.

Oster E. Unobservable Selection and Coefficient Stability: Theory and Evidence. Journal of Business and Economic Statistics. 2019; 37(2): 187-204.

---

## [Decision Letter · Decision Letter 1]

15 May 2023

PONE-D-22-15238R1THE SHADOW OF THE FAMILY: HISTORICAL ROOTS OF SOCIAL TRUST IN EUROPEPLOS ONE

Dear Dr. Kravtsova,

Thank you for submitting your revised manuscript. The reviewer and I agree that the manuscript has improved much. Yet, there are still some open points that should be addressed. Therefore, we invite you to submit a revised version of the manuscript that addresses the points raised during the review process.

Two reviewers think that the revised article is ready for publication, although one of them asks for the article first being proof-read by a native speaker to eliminate language mistakes. A third reviewer is not yet convinced by some of the authors' reactions. The reviewer provides some detailed suggestions for changes and beyond that asks the authors to address the potential problem of non-representative subnational data. I agree that the authors' response to this issue could have been more detailed and transparent. The authors primarily add footnote 10 to the manuscript, the statements in which might be difficult to comprehend for many readers. What do the authors mean by saying they are not "constructing indicators" at the subnational level and why does that make a difference? Can all readers be expected to know what primary sampling units are? - a term the authors use repeatedly without explaining it. The authors should at least make a clear statement whether unrepresentative data can have consequences for the internal or external validity of results, rather than a non-scientific statement like "this is not a crucial issue". The reviewer makes more suggestions how the issue could be addressed. The authors should evaluate and answer in their response which of these suggestions are useful to make their analysis more convincing or make its limitations transparent. Beyond the reviewer's comments, I would ask the authors to fix equation one. Also (vectors of) control variables are accompanied by (vectors of) parameters to be estimated. I don't understand why the authors speak in this particular spot of "historical countries(sic!) fixed effects". These are simply country fixed effects, right? The manuscript still includes many mistakes. "e.g." includes two punctuation marks and is regularly followed by a comma. One weblink in footnote 8 includes a space. Lateral is not capitalized. I see at least three different citation styles used in-text. "in Appendix" requires an article. What do the authors mean by "Albania, Croatia, France, Hungary, Romania, Slovakia, Sweden, and United Kingdom (which correspond to five historical states: Albania, France, Great Britain and Wales, Hungary, Romania, Scotland, and Sweden)."? And how do they use the term Great Britain (especially w.r.t. Scotland and Wales, which appear to be separate)? These are just some quick and random observations (i.e., not a complete list) that underline that the manuscript needs more work, as was also underlined by one of the reviewers who suggested a professional proof-reader.

We look forward to receiving your revised manuscript.

Kind regards,

Jerg Gutmann

Academic Editor

PLOS ONE

Journal Requirements:

Reviewers' comments:

Reviewer's Responses to Questions

**Comments to the Author**

1. If the authors have adequately addressed your comments raised in a previous round of review and you feel that this manuscript is now acceptable for publication, you may indicate that here to bypass the “Comments to the Author” section, enter your conflict of interest statement in the “Confidential to Editor” section, and submit your "Accept" recommendation.

Reviewer #2: (No Response)

Reviewer #3: (No Response)

Reviewer #4: All comments have been addressed

2. Is the manuscript technically sound, and do the data support the conclusions?

Reviewer #2: Partly

Reviewer #3: Yes

Reviewer #4: Yes

3. Has the statistical analysis been performed appropriately and rigorously? 

Reviewer #2: N/A

Reviewer #3: Yes

Reviewer #4: Yes

4. Have the authors made all data underlying the findings in their manuscript fully available?

Reviewer #2: Yes

Reviewer #3: (No Response)

Reviewer #4: Yes

5. Is the manuscript presented in an intelligible fashion and written in standard English?

Reviewer #2: Yes

Reviewer #3: Yes

Reviewer #4: Yes

6. Review Comments to the Author

Reviewer #2: While I recognize some effort in the revision, I found the paper harder to read, this time. Somehow I have the impression the authors just want to push their claims with too much vehemence. But the statistical analysis, although implemented with rigour is not able to lift all doubts regarding those claims. This has to do with data typology, number of observations and the complexity of the subject and time frame investigated.

I am in favour of showing this kind of work, but I would welcome a bit more humility in the exposition and the claims made, especially when the subject is easily transposed to larger levels e.g. formal and informal institutions. I also strongly advise to pay more attention to the limitations that an analysis like this is clearly confronted with, in order to tease out the most reliable evidence.

In the attached file you will find some more comments (in capital when included in the previous reply to reviewer text) and at the end of the previous exchange I have added further points.

Reviewer #3: (No Response)

Reviewer #4: I am basically happy with all modifications added as a consequence of the reviews but would like to encourage the authors to give the ms to a native speaker before the final submission to the journal

7. PLOS authors have the option to publish the peer review history of their article (what does this mean?). If published, this will include your full peer review and any attached files.

Reviewer #2: No

Reviewer #3: No

Reviewer #4: No

---

## [Author Response · Author response to Decision Letter 1]

14 Aug 2023

Dear Editor,

We are very grateful for having been given the opportunity to revise and improve the paper further. We did our best to address remaining comments and made a professional proof-reading of the text by the native speaker. Below we present our responses to the comments. We start with the editorial comments and then present our responses to the reviewer’s comments. All comments are written in italics, followed by our responses written in normal font.

Editorial comments

The authors primarily add footnote 10 to the manuscript, the statements in which might be difficult to comprehend for many readers. What do the authors mean by saying they are not "constructing indicators" at the subnational level and why does that make a difference? Can all readers be expected to know what primary sampling units are? - a term the authors use repeatedly without explaining it. The authors should at least make a clear statement whether unrepresentative data can have consequences for the internal or external validity of results, rather than a non-scientific statement like "this is not a crucial issue". The reviewer makes more suggestions how the issue could be addressed. The authors should evaluate and answer in their response which of these suggestions are useful to make their analysis more convincing or make its limitations transparent.

We agree that those statements were ambiguous and we are grateful for the opportunity to clarify our argumentation. By the phrase “we don’t use LiTS data to construct any indicators at the subnational level” we wanted to emphasize that our study is not aimed to measure mean levels of trust at the subnational level. In other words, we are not interested in measuring the level of trust in PSU #1 or in PSU #24 or in any other particular PSU (Please, note that wherever possible LiTS uses local electoral units as PSU. In regions where local electoral territorial units were not available, Census Enumeration Districts or local authorities were used as PSUs. This clarification is added to the text.) If we would be interested in such measurements, then the non-representativeness of LiTS at subnational level, indeed, would be a crucial limitation. 

Rather, in our paper we measure indicators trust of individuals living within PSUs and regress these indicators on historical family indicators at the PSU level. Our interest is to estimate the impact of historical family indicators on individual trust. So, the question is whether the non-representativeness of LiTS at the PSU level compromises our econometric findings, i.e., whether it causes any bias in the OLS estimates of the impact of historical family indicators on individual trust. We believe that it doesn’t cause any bias, and here’s why. 

First of all, let us formulate why there may be a problem at all. The non-representativeness of LiTS at the PSU level means that the composition of the PSU samples differs from the composition of the corresponding PSU populations. This means that our dependent variable is, probably, measured with errors, as we would observe different values of trust within each PSU if PSU samples were representative. Therefore, the issue of non-representativeness is reduced to the issue of measurement errors in the dependent variable.

Next, as it is well known, measurement errors in the dependent variable bias the OLS estimator only if they are non-classical, i.e., if they are correlated with the explanatory variable (in this case the explanatory variable will be endogenous). If errors are not correlated with the explanatory variable, they are classified as classical and can be viewed just as a noise which doesn’t bias the OLS estimator [see the detailed review by Bound, Brown & Mathiowetz (2001) in Chapter 59 of the Handbook of Econometrics edited by J. Heckman and E. Leamer] Therefore, the key question is whether errors in our trust variable are classical or not.

To answer this question, we refer to the sampling procedure used in LiTS. It consists of two stages. First, for each country 50-75 PSUs are chosen from a larger country-specific list of PSUs with probability of selection proportional to the PSU size. Second, households which are surveyed are randomly chosen within each PSU. The random sampling of households within a PSU means that even if the sample doesn’t represent the proportions of the general population of PSU and trust values of the sampled individuals differ from the trust values of the population, these differences should be considered as random and thus not be correlated with any factors. In sum, possible measurement errors in our dependent variable due to the non-representativeness of LiTS at the subnational level should not bias our econometric estimates as those errors are random (by survey design). 

Having said that, we acknowledge that possible concerns about the non-representativeness of LiTS’ at the regional level still may exist. Therefore, we made the effort to provide robustness checks to ease those concerns. First of all, we carefully considered the reviewer’s suggestion to look at the correlation between the dependent variable (trust) and historical family indicators at the country level (see our reply to the reviewer’s comments). Secondly, we provided an alternative robustness check, relying on external data source. We used “Values in the Russian Regions” survey (Almakaeva et al., 2019), which is representative at the regional level and uses exactly the same questions on trust as LiTS does. It appeared that the regional-level correlation between the out-group trust measure from LiTS and the out-group trust measure from that Russian survey is about 0.83***, which suggests that LiTS data provide rather good estimates for the mean levels of out-group trust at the subnational level, at least for one country, Russia. We add to the revised version of the paper a paragraph where we provide all our arguments and mention this robustness check (lines 456-470).

Concerning internal validity, the argumentation presented above implies that our findings are valid internally (as our research design and analysis answer the research question without bias). 

Beyond the reviewer's comments, I would ask the authors to fix equation one. Also (vectors of) control variables are accompanied by (vectors of) parameters to be estimated. I don't understand why the authors speak in this particular spot of "historical countries(sic!) fixed effects". These are simply country fixed effects, right? 

Yes, these are country fixed effects. However, historical and contemporary countries differ to some extent. For example, in the historical census lists Scotland is mentioned separately from England and Wales. The Hungarian part of the Habsburger empire included the present-day territories of Croatia and Slovakia. Therefore, historical countries FE differ from (contemporary) countries FE. We include FE for historical countries but not FE for the contemporary countries to better control for omitted variables which may affect both historical family structures and out-group trust. Any contemporary societal characteristics, which may be omitted for the model, don’t affect historical family structures but rather may be influenced by them.

The manuscript still includes many mistakes. "e.g." includes two punctuation marks and is regularly followed by a comma.

This problem was fixed. 

One weblink in footnote 8 includes a space.

The space was removed.

Lateral is not capitalized.

We use capital letters when Lateral means the variable name, while lower-case is used when lateral is an adjective (lateral extensions). We double-checked this. 

"in Appendix" requires an article.

We fixed that. 

 I see at least three different citation styles used in-text.

Now we stick to one citation style.

What do the authors mean by "Albania, Croatia, France, Hungary, Romania, Slovakia, Sweden, and United Kingdom (which correspond to five historical states: Albania, France, Great Britain and Wales, Hungary, Romania, Scotland, and Sweden)."? And how do they use the term Great Britain (especially w.r.t. Scotland and Wales, which appear to be separate)?

As we mentioned earlier, historical and contemporary states do not entirely overlap. Contemporary Croatia, Hungary and Slovakia were part of the Habsburger empire (Hungarian part). In the 19th century England and Wales had a separate census from Scotland. Great Britain was just a misprint, we meant England. We corrected it in the revised version and used “the Hungarian part of the Habsburger empire” instead of historical Hungary (see lines 83-84).

Reviewer’s comments

Old comments

Reviewer’s comment 1.1 (original)

The Life in Transition Survey is representative at the national level, but the authors use single observations also at lower levels of aggregation. I do not think this is a correct procedure, in terms of statistics. If the authors used any additional computation to overcome such sample design problems or if they are in possession of further details that allow treating the data as statistically representative at the subnational level, they must provide details on it. Otherwise, the reader remains under the impression that it is not viable to use the LITs data at the subnational level at all. 

Our response 1.1 (previous)

We agree that this is a methodological limitation of our study, which we now mention in the revised version (see footnote 10 at p.19). However, we believe that the non- representativeness of LiTS at the subnational level it is not a crucial issue in our case as we don’t use LiTS data to construct any indicators at the subnational level. Rather, we assign historical indicators taken from external data sources to all individuals living within the same primary sampling units (PSU). In this case, a two-stage quasi-random sampling procedure used in LiTS (first, for each country 50-75 PSUs are chosen from a larger country-specific list of PSUs with probability of selection proportional to the PSU size, and, second, households are randomly chosen within each PSU) should prevent any systematic bias in the composition of PSU samples and thus should guarantee that our estimated coefficients are free of the sample-selection bias. The representativeness of LiTS at subnational level would be a pleasant bonus in this case, but it shouldn’t affect the econometric estimates.

Additionally, we would like to note that the literature contains plenty published examples which follow a similar procedure as we do: first, match subnational level indicators to microdata from cross-national surveys which are not representative at the subnational level, and, second, estimate the impact of those indicators on some individual level dependent variables. For instance, Ajzenman et al., (2022) in their recent study also use LiTS data to show that the exposure to mass transit migration measured at the PSU level (the distance from the locality to the closest transit migrant route) affects individual entrepreneurial activity and anti-migration sentiments. Schulz et al., (2019) assigns historical indicators calculated at the level of contemporary subnational regions to the European Social Survey (ESS) data which are not representative at the subnational level as well. Duranton et al. (2009) derive their measures of social capital and income inequality at the NUTS-2 level from the European Community Household Panel (ECHP) microdata, which are not representative at the subnational level. This last example, probably, is vulnerable to the problem of the non-representativeness at the subnational level, but this is not the case of our paper. 

Reviewer’s comment 1.2 (new)

I do not think the argumentation is satisfying. Just because other authors “got away” with a wrong research design does not justify not taking care of it. I do think the following is still necessary:

- some more findings, which could take the form of a robustness test in which you e.g. correlate historical subnational data with national present trust-values, or some more detailed descriptive statistics such as a map of the dependent variable at the subnational level that can be compared to your already produced map with historical data. You can opt for a graph that visually pictures the correlations given the number of observations of countries is low

- such findings should be commented in order to assess whether the lack of representativty at the subnational level of the lis design risks altering any of your findings. It is important to do so as the dependent variable relies on that sample.

- if any of your findings change, please make sure you report the issue also in the section dedicated to the limitations of the study

Please note that your figure 6 right now has a note that stresses within-country variability as key factor for your findings. This stresses the need for greater precision in dealing with this important limit of the data. I recommend revising that note, by the way because it currently does not make too much sense but reports a contradictory sense.

Our response 1.2 (new)

We accept that our argumentation was unclear and non-convincing. Let us explain better and in more details why we believe that the non-representativeness of LiTS at the subnational (PSU) level should not compromise our econometric findings. 

First of all, let us formulate why there may be a problem at all. The non-representativeness of LiTS at the PSU level means that the composition of the PSU samples differs from the composition of the corresponding PSU populations. This means that our dependent variable is, probably, measured with errors, as we would observe different values of trust within each PSU if PSU samples were representative. Therefore, the issue of non-representativeness is reduced to the issue of measurement errors in the dependent variable.

Next, as it is well known, measurement errors in the dependent variable bias the OLS estimator only if they are non-classical, i.e., if they are correlated with the explanatory variable (in this case the explanatory variable will be endogenous). If errors are not correlated with the explanatory variable, they are classified as classical and can be viewed just as a noise which doesn’t bias the OLS estimator [see the detailed review by Bound, Brown & Mathiowetz (2001) in Chapter 59 of the Handbook of Econometrics edited by J. Heckman and E. Leamer] Therefore, the key question is whether errors in our trust variable are classical or not.

To answer this question, we refer to the sampling procedure used in LiTS. It consists of two stages. First, for each country 50-75 PSUs are chosen from a larger country-specific list of PSUs with probability of selection proportional to the PSU size. Second, households which are surveyed are randomly chosen within each PSU. The random sampling of households within a PSU means that even if the sample doesn’t represent the proportions of the general population of PSU and trust values of the sampled individuals differ from the trust values of the population, these differences should be considered as random and thus not be correlated with any factors. In sum, possible measurement errors in our dependent variable due to the non-representativeness of LiTS at the subnational level should not bias our econometric estimates as those errors are random (by survey design). 

We very much hope that our argumentation is more clear now. We believe that this also explains the widespread use of a research design similar to ours in many quantitative studies. 

Having said that, we acknowledge that possible concerns about the non-representativeness of LiTS’ at the regional level still may exist. Therefore, we made the effort to provide robustness checks to ease those concerns. 

First of all, we carefully considered the reviewer’s suggestion to look at the correlation between the dependent variable (trust) and historical family indicators at the country level. If we understand well, the idea behind that suggestion is that that correlation could indicate what the correlation at the subnational level could be if LiTS would be representative at the subnational level (since LiTS is representative at the national level). While we like this idea, unfortunately, it doesn’t work in practice. First, we have only eight country-level observations, which makes any estimates very imprecise. Second, and more important, countries are too wide and heterogeneous units. Averaging trust and historical family indicators erases almost all meaningful variation in these variables. The variation that we use for identification in our paper is subnational and comes from within countries (as we include country FE). Therefore, we are afraid that the correlation obtained at the country level hardly may be indicative of the correlation at the subnational level.

While trying to invent alternative checks, we turned to the survey “Values in the Russian Regions” which was conducted in January–February 2019 (Almakaeva et al., 2019). This survey is representative at the regional level and uses exactly the same questions on trust as LiTS does (these data will be available as a supplementary material to our paper). It appears that the regional-level correlation between the out-group trust measure from LiTS and the out-group trust measure from that Russian survey is about 0.83***, which suggests that LiTS data provide rather good estimates for the mean levels of out-group trust at the subnational level, at least for one country, Russia. We mention this check in the revised version of the paper.

We add to the revised version of the paper a paragraph where we provide all our arguments (lines 456-470).

Concerning Fig.6, we revised that note. However, we cannot eliminate it completely as it explains to the readers how Fig. 6 is constructed. It reflects the within-country variation, as exactly that variation is used in our regressions for identification (as historical country fixed effects which remove all between-country variation are included in our regressions). 

Comment 2.1 (original)

the deepest explanation of isolative vs. cooperative comes only in the conclusion. Better treatment of the point should be anticipated in the manuscript. 

Response 2.1 (new)

In the revised version, we introduce the explanation of the “isolative” and “cooperative” components in the introduction and then we elaborate it in the hypotheses section.

Comment 2.2 (new)

I saw the new text you included but still think it remains vague. Please make an effort to improve it. Line 72-73 “isolative” is still unclear as it seems to describe also horizontal and not vertical – please clarify this a little bit more. My suggestion is to rework the paragraph starting at line 195. Can you summarize this logic in a diagram? and refer back to it in your conclusions? The point seems to be relevant for your findings but the exposition in the paper currently does not allow to properly grasp the implications.

Response 2.2. (new)

You are completely right: both horizontal and vertical family extensions mean a larger number of relatives and lead to more self-sufficiency of the household and less need for social interactions beyond the kin group. We tried to make it more clear in the revised version (see line 62). Additionally, we reworked that paragraph and added a special Fig.1 “Theoretical mechanisms of the effect of family extendedness on out-group trust.”, where we illustrate the negative and the positive pathways running from family extendedness to out-group trust. We refer to the “isolative” and “cooperative” components of family extendedness in the concluding section.

Fresh comments

line 129 “gender equality stimulated…” are these just co-evolving factors or has causality been proved here? If yes, please cite relevant literature.

We agree that the causal impact of gender inequality on other variables hasn’t been proved (as it is extremely difficult to find a valid instrument), this we corrected that sentence. 

167-171: argumentation is VERY speculative.

We revised this paragraph (see lines 171-183). We believe that persistence of family structure is not the only institutional channel supporting persistence of trust. There might be persistence or path-dependence of other societal institutions that may have an effect on trust like good government, effective law, or participatory institutions. All these institutions provoke social trust by stimulating cooperation between people and making it less risky. Consequently, trust persistence may rely on the continuity of these institutions.

Is Putnam’s argument regarding horizontality among peers really isomorphically valid also at the family-level? 

We believe that Putnam’s argument is applicable to the family level as well. It is even more important to elaborate impartial norms of interaction between relatives of equal social status within a household because otherwise they would not be able sustain regular daily communications which they cannot avoid while living under one roof. We added this argumentation to the text, see lines 215-218 in the revised version. 

Lines 202-205 simply extend the argument of trusting each other in similar conditions (blood-link) to ones in which the blood-link is missing.

We provided a link to the paper by Welzel and Delhey (2015) where the authors state that out-group trust grows to a large extent out from in-group trust because people should learn to trust someone before they can trust everyone. Their theory justifies the extension of our argument of trusting each other within the family to trusting strangers. See lines 221-223 in the revised text.

How about mentioning the role that the share of servants in the family plays for your key argument? 

Thank you for this suggestion. The presence of servants in the household is responsible for communications with the out-groups on a daily basis and for the formation of mutually acceptable social norms. We added this paragraph to the text, see lines 224-228.

Can you please add the correlations of factors with the latent factors you extrapolate from the PCA? that is relevant to see as a reader.

We added Table A4 (in the Appendix) with the results of two separate Principal Components Analyses that we performed to create generational and gender hierarchy indices. Numbers indicate the strength of correlation of each variable with the eigenvector of the first principal component for each PCA.

When you mention that a higher share of female-headed hh stands for greater independence, please consider that this can be a phenomenon due to war or specific migratory patterns.

We agree with this comment and made a robustness check. We re-constructed the gender hierarchy index without this item and run the regressions with such an index. Table A5 in Appendix shows that the results do not change. 

I found it rather odd to discuss hunter-gatherers in the context of European data in the last centuries. While I think it is ok to include the variable stressing some kind of geographical determinants on culture, I think the current discussion around the variable in the text is misplaced.

Thank you for this remark. We tried to clarify this issue in the revised text (see lines 440-446). Welzel et al., (2021) stress the fact that territories well suited for hunting and gathering adopted agriculture several thousand years later than the first adopters in the Middle East. According to their theory, it created favorable conditions for the future Industrial Revolution and left an imprint on contemporary social values and institutions. In light of this evidence we consider it is appropriate to use land suitability for hunting and gathering as a control variable.

Figures 4 and 5: you need to exclude the outlier Albania at least to give a better perspective on the results you find. In this moment the two figures are not intelligible because of that outlier. You can include either of the two versions in the appendix.

We provided these figures without Albania in Appendix (see Fig. A4 and Fig. A5). 

I think you need to be much more precise in data labelling throughout. Which variables are historical which ones are current? E.g. in figures 4 and 5 it is not clear.

We changed the variable labels in Figures 4 and 5 so that it is now clear that we are using historical data. 

Lines 8-10: you start with “evidently…” but there could by many other reasons for such finding.

Unfortunately, we didn’t find “evidently…” in those lines.

524-525: Should’nt it say “decreases”?

Sure, it means decreases. We corrected this.

The limitations section is ok but rather hidden in the text. I suggest to expand it and to provide a title for it.

Following your suggestion, we expanded this section and provided a title for it.

While I recognize some effort in the revision, I found the paper harder to read, this time. Somehow I have the impression the authors just want to push their claims with too much vehemence. But the statistical analysis, although implemented with rigour is not able to lift all doubts regarding those claims. This has to do with data typology, number of observations and the complexity of the subject and time frame investigated.

I am in favour of showing this kind of work, but I would welcome a bit more humility in the exposition and the claims made, especially when the subject is easily transposed to larger levels e.g. formal and informal institutions. I also strongly advise to pay more attention to the limitations that an analysis like this is clearly confronted with, in order to tease out the most reliable evidence.

Following your comments we tried to make the limitations of our study more transparent to readers and smoothed some claims made. We spent a lot of effort and did our best to provide a validation test for the subnational LiTS data (see the response to your previous comments). Unfortunately, this limitation still remains and we mention it in our limitation section.

---

## [Decision Letter · Decision Letter 2]

30 Nov 2023

THE SHADOW OF THE FAMILY: HISTORICAL ROOTS OF SOCIAL TRUST IN EUROPE

PONE-D-22-15238R2

Dear Dr. Kravtsova,

We’re pleased to inform you that your manuscript has been judged scientifically suitable for publication and will be formally accepted for publication once it meets all outstanding technical requirements.

Kind regards,

Jerg Gutmann

Academic Editor

PLOS ONE

Additional Editor Comments (optional):

Reviewers' comments:

Reviewer's Responses to Questions

**Comments to the Author**

1. If the authors have adequately addressed your comments raised in a previous round of review and you feel that this manuscript is now acceptable for publication, you may indicate that here to bypass the “Comments to the Author” section, enter your conflict of interest statement in the “Confidential to Editor” section, and submit your "Accept" recommendation.

Reviewer #2: All comments have been addressed

2. Is the manuscript technically sound, and do the data support the conclusions?

Reviewer #2: Yes

3. Has the statistical analysis been performed appropriately and rigorously? 

Reviewer #2: Yes

4. Have the authors made all data underlying the findings in their manuscript fully available?

Reviewer #2: Yes

5. Is the manuscript presented in an intelligible fashion and written in standard English?

Reviewer #2: Yes

6. Review Comments to the Author

Reviewer #2: Thank you for responding to my concerns. I believe the paper is in a better shape now, good luck with your future work.

7. PLOS authors have the option to publish the peer review history of their article (what does this mean?). If published, this will include your full peer review and any attached files.

Reviewer #2: No

---

## [Editor Report · Acceptance letter]

31 Jan 2024

PONE-D-22-15238R2 

PLOS ONE

Dear Dr. Kravtsova, 

I'm pleased to inform you that your manuscript has been deemed suitable for publication in PLOS ONE. Congratulations! Your manuscript is now being handed over to our production team.

Kind regards, 

on behalf of

Prof. Dr. Jerg Gutmann 

Academic Editor

PLOS ONE